# Andean mountain building and magmatic arc migration driven by subduction-induced whole mantle flow

W.P. Schellart [1,2]

Subduction along the western margin of South America has been active since the Jurassic, but Andean orogeny started in the middle Cretaceous and was preceded by backarc extension in the Jurassic-Early Cretaceous. The timing and sequence of these events has remained unexplained. Here I present a four-dimensional buoyancy-driven whole-mantle subduction model implying that the ~200 Myr geological evolution can be attributed to sinking of a wide slab into a layered mantle, where upper-mantle wide-slab subduction causes backarc extension, while whole-mantle (upper+lower) wide-slab subduction drives Andean orogeny. The model reproduces the maximum shortening and crustal thickness observed in the Central Andes and their progressive northward and southward decrease. The subduction evolution coincides with a 29° decrease in slab dip angle, explaining ~200 km of Jurassic-present eastward migration of the Central Andean magmatic arc. Such arc migration negates proposed long-term subduction erosion and continental destruction, but is consistent with long-term crustal growth.

[1] Department of Earth Sciences, Vrije Universiteit Amsterdam, 1081 HV Amsterdam, The Netherlands. [2] School of Earth, Atmosphere & Environment, Monash University, Melbourne VIC 3800, Australia. Correspondence and requests for materials should be addressed to W.P.S. (email: w.p.schellart@vu.nl)

Mountain building at convergent margins is generally attributed to the collision of two continents[1–3] or the accretion of continental ribbons, oceanic plateaus or arc terranes to the overriding plate of a subduction system[4–6]. In this respect the Andes are unique, being located at the South American subduction zone (Fig. 1), which has not experienced any collision or accretion event in the last ~200 Myr[7–9] except for local Late Cretaceous-Palaeocene accretion in the northern Andes[10]. Andean mountain building and shortening have been active since the middle to Late Cretaceous (~120–70 Ma)[7, 11–15] during several orogenic phases, but were preceded by a period of Jurassic-Early Cretaceous backarc extension[7, 9, 13, 14] that was most pronounced in the north and south, and less extensive in the centre[9]. Other unique traits of the Andes include its central curved geometry, most notably the Bolivian orocline in the centre[16], and the variation in trench-normal shortening and crustal thickness along the Andes (Fig. 1d, e), with maximum shortening and crustal thickness in the Central Andes, progressively decreasing northward and southward[16–20]. In addition, the cordilleran mountain range is characterized by magmatic arc migration and widening in the Central Andes since the Jurassic/Early Cretaceous[8, 11] (Fig. 1b, c), but a lack thereof in the north and south (Fig. 1a).

Different models have been proposed to explain Andean mountain building, ascribing it to subducting plate age[21, 22] or overriding plate velocity[23, 24]. Explanations for trench-parallel variation in Andean shortening[25] and topography[22] call on trench-parallel variations in trench sediment thickness or subducting plate age. Global statistical analyses show, however, that age and trench sediment thickness are not correlated with overriding plate deformation rate[26], while overriding plate velocity cannot explain the trench-parallel variation in shortening. In addition, such models do not explain the earlier extension phase nor the migration and widening of the arc. Cordilleran mountain building has also been ascribed to the subduction of aseismic ridges or plateaus in conjunction with flat slab subduction[27, 28], but observational constraints[29], reconstructions[30], geodynamic models[31, 32] and statistical analysis[26] indicate that if such buoyant features produce permanent deformation in the overriding plate in the form of shortening, this deformation is only of local character. More generally, overriding plate deformation at subduction zones (both extension and shortening) has been ascribed to subduction-induced flow in the sub-lithospheric mantle[33–39]. How such mantle flow might have been responsible for early extension in the Jurassic-Early Cretaceous and later cordilleran mountain building in the Andes since the ~middle Cretaceous will be tested in this contribution.

Here I present a dynamic, buoyancy-driven, whole-mantle numerical subduction model to test a hypothesis in which the formation of the Andes is mostly a consequence of the long-term (~200 Myr) progressive evolution and large width (trench-parallel extent) of the subduction zone, focusing on the role of subduction-induced mantle flow in driving overriding plate deformation. The numerical experiment uses the code Underworld[6, 40–43] (Methods section) and builds on previous generic subduction models[37] to simulate time-evolving subduction of a 6000 km wide (trench-parallel extent) oceanic plate below a continental plate in a very large three-dimensional layered whole-mantle domain (comparable to the Nazca-South America subduction setting). The numerical geodynamic model reproduces and explains several primary characteristics of the Andes, including the variation in shortening and crustal thickness along the Andes, its early extension phase, the curvature of the orocline and trench, and the migration of the magmatic arc. It thereby demonstrates the role and importance of subduction zone size, time and subduction depth in the formation and evolution of cordilleran mountain belts.

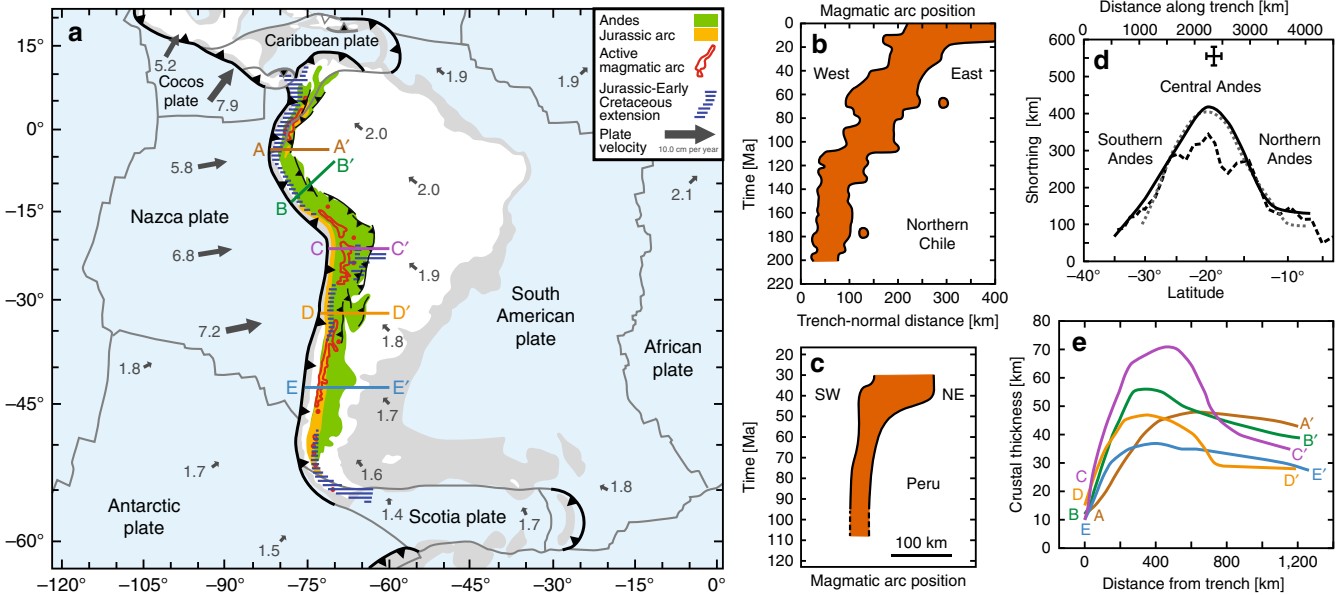

**Fig. 1** Tectonics and geology of the Andes region. **a** Tectonic map showing the current plate velocities, the Andes mountain belt, Jurassic-Early Cretaceous extension in the Andean region[9], the Jurassic magmatic arc (Methods section) and the present magmatic arc. Note that the five transects (A-A', B-B', C-C', D-D', E-E') are shown in **e**. **b**, **c** Evolution of the magmatic arc position in Northern Chile (21°–26°S, based on Scheuber et al.[8]) and Peru (simplified from Jaillard and Soler[11]). **d** Trench-parallel variation in Andean shortening. Continuous black line is based on balanced cross-sections and Eocene-present tectonic reconstruction[19], dotted grey line is based on Eocene-present tectonic reconstruction[18], dashed black line is based on crustal thickness (assuming an original thickness of 40 km)[17], and Central Andes data point is based on structural mapping of Late Cretaceous-present deformation[12]. **e** Profiles showing the crustal thickness in five locations (see **a**) along the Andes (derived from the crustal thickness map of Chulick et al.[20])

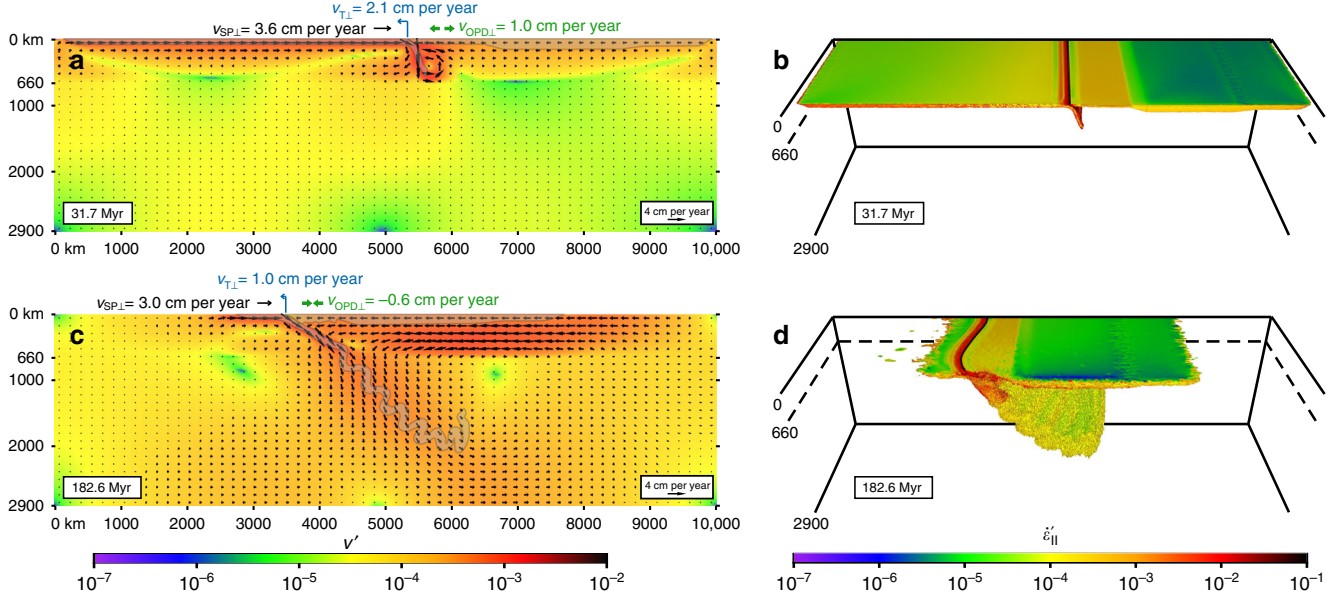

**Fig. 2** Model results showing central cross-sections and 3D views of the subduction model simulating Nazca/Farallon oceanic plate subduction below the South American plate. **a, b** Upper mantle subduction stage (31.7 Myr, corresponding to the Middle Jurassic). **c, d** Whole-mantle subduction stage (182.6 Myr, corresponding to the Late Cenozoic). Cross-sections (**a, c**) show velocity field and magnitude (vectors and colour scale; $v'$ is the non-dimensional velocity), while transparent grey zones with dark grey outline show subducting plate-slab geometry and overriding plate geometry. 3D views (**b, d**) show non-dimensional second invariant of the strain rate in the plates and slab

## Results

### Subduction and mantle flow and overriding plate deformation.

During the initial stage of subduction, the slab sinks and rolls back in the upper mantle (Figs. 2a, b and 3a), slow in the centre and more rapidly near the edges, and induces a return flow, dominantly poloidal and with an additional toroidal component near the lateral edges (Fig. 4a). Subduction is accommodated by trench retreat and trenchward subducting plate motion (Fig. 3a), with the latter component dominating as expected for this wide subduction zone setting[35]. The overriding plate experiences minor forearc deformation and significant backarc extension up to 1.3 cm per year in the centre amounting to 247 km of extension at 67.5 Myr (Fig. 3a, b). Extension results from basal tractions induced by slab rollback-driven upper mantle return flow (Figs. 2a, 4a and 5a–d), with the greatest extension near the lateral slab edges due to the additional toroidal flow, with a maximum of 463 km at 2400 km from the centre (Fig. 3b).

Such behaviour is consistent with observations of Jurassic-Early Cretaceous backarc extension in southernmost (Rocas Verdes Basin) and northernmost (Colombian Marginal Seaway) South America, but reduced backarc extension in the centre[9]. The amount of extension along South America's west coast has not been quantified, but the Rocas Verdes Basin and Colombian Marginal Seaway both developed backarc spreading as implied by remnants of backarc oceanic crust in the southern and northern Andes[9, 44, 45]. Backarc spreading in an ocean-continent subduction setting generally implies several hundred km of backarc opening, which is of the same order of magnitude as observed in the numerical model. Extension in the central Andes region was likely an order of magnitude less compared to the north and south as implied by the lack of backarc oceanic crust and the limited extensional structures and sedimentary rocks, and so probably an order of magnitude less than the 247 km of extension observed in the model. This discrepancy likely results from the adopted initial overriding plate set-up in the model, with a relatively thin forearc-backarc region and weak backarc rheology (Fig. 6). A stronger backarc, for example an order of magnitude

higher viscosity, would be plausible for the first few tens of million years following subduction initiation, and would reduce the backarc extension rate and amount in the centre by an order of magnitude, more in line with the sparse amount of observations. The longer period of extension in the north and south, accompanied with progressive thermal weakening of the backarc, would still allow for major backarc extension in these regions.

From ~33-35 Myr until ~48 Myr, as the slab approaches and touches the 660 km discontinuity, plate and trench velocities decrease (Fig. 3a) and the frontal slab part forms a horizontal segment at the discontinuity. This is followed by the formation of slab folds that form at the discontinuity, which progressively sink into the lower mantle forming a folded slab pile (Fig. 2c, d). The periodic slab folding causes periodic changes in subducting plate and trench velocities (Fig. 3a). Once the first slab fold enters the lower mantle at ~67.5 Myr, backarc extension changes to backarc shortening in the centre (Fig. 3a), while extension continues periodically until 80–120 Myr away from the centre (Fig. 3b), which is consistent with observations from the Neuquén Basin in the Southern Andes[14]. In general, the change from backarc extension to shortening at ~67.5–120 Myr in the geodynamic model (representing ~132.5–80 Ma in nature) is consistent with the onset of shortening in the Andes at ~120–70 Ma. During shortening, the backarc deformation rate shows periodic to episodic behaviour, consistent with earlier proposals of periodic deformation in the Central Andes based on observations[11, 46], with a maximum shortening rate of 0.9 cm per year. The average rate of shortening increases with time until ~140 Myr, after which it remains more constant. Finite shortening (~ 300 km from 0 to 182.6 Myr and ~ 550 km from 67.5 to 182.6 Myr) is maximum in the centre (cf. Central Andes) and decreases towards the north and south (cf. northern and southern Andes) (Fig. 3c, d), which is consistent with observations of overriding plate shortening[16–19, 47].

A particularly good fit between model and nature is obtained when ~200 km of western cordillera shortening[12], here postulated to have occurred along all of the western margin of South

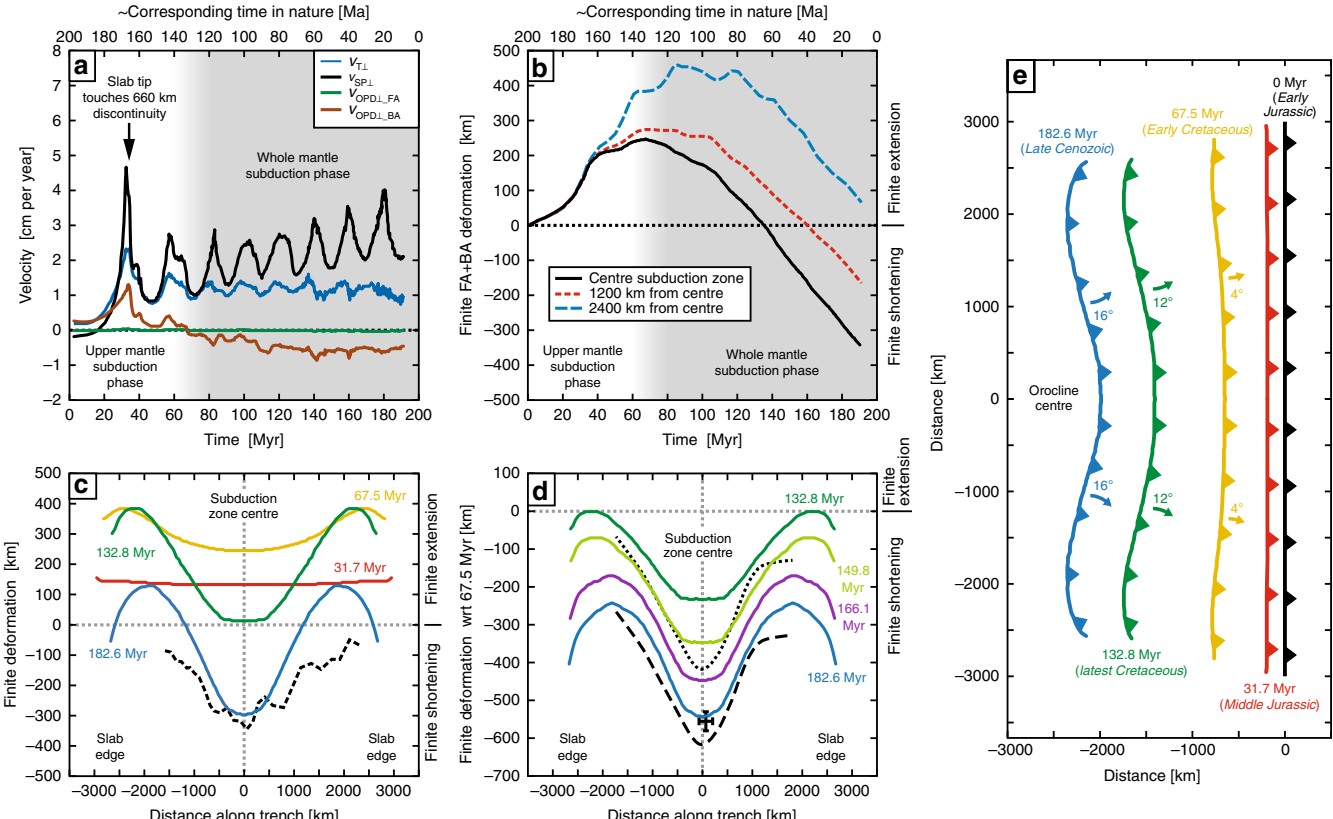

**Fig. 3** Numerical results showing the temporal evolution of subduction zone kinematics and overriding plate deformation and trench geometry. **a** Trench-normal trench velocity ($v_{T\perp}$, retreat is positive), subducting plate velocity ($v_{SP\perp}$, trenchward is positive), forearc deformation rate ($v_{OPD-FA\perp}$, extension is positive), and backarc deformation rate ($v_{OPD-BA\perp}$, extension is positive) in the subduction zone centre. **b** Trench-normal finite deformation of forearc and backarc in three locations (extension is positive, shortening is negative). Note that the slope of the curves gives the (instantaneous) deformation rate, with a positive slope indicating an extension rate, and a negative slope indicating a shortening rate. **c** Variation of trench-normal overriding plate finite deformation along the trench with respect to 0 Myr (start of model). Black dashed line gives estimated trench-parallel variation in Andean shortening based on crustal thickness[17]. **d** Variation of trench-normal overriding plate finite deformation along the trench with respect to 67.5 Myr (~end of extension phase in centre of subduction zone). Black dotted line gives estimated trench-parallel variation in Andean shortening based on balanced cross-sections and reconstructions[19]. Black dashed line is based on black dotted line but includes ~200 km of additional shortening as documented in the western cordillera[12], which is postulated to exist all along the coast. Central Andes data point is also plotted[12]. **e** Evolution of the trench geometry and position. Curved arrows indicate rotation sense and numbers give maximum rotation of the orogen on either side of the orocline centre. Please note that the colour coding in (**c**–**e**) is related (except for the black lines)

America, is added to the curve for Eocene–present shortening (cf. dashed black curve and blue curve in Fig. 3d). The increased shortening close to the lateral slab edges in the model could possibly represent the underthrusting/subduction of the backarc oceanic basins of the Rocas Verdes Basin and Colombian Marginal Seaway, thereby accommodating several hundred kilometres of shortening.

**Mechanism of overriding plate deformation.** During the early extension phase, subduction is dominated by slab rollback and the induced return flow is confined to the upper mantle. The trench-normal dimension of the return flow scales with the upper mantle depth and is therefore short (~700 km), dragging the frontal part of the overriding plate away from the remainder of the overriding plate towards the retreating subduction zone hinge (Figs. 2a, 4a and 5a–d). This causes deviatoric tension and extension in the overriding plate (Figs. 3a, b and 7a–c, 31.7 Myr), with maximum deviatoric tensile stresses of 20–60 MPa occurring within 1200 km from the trench. In contrast, during the following shortening phase, subduction is dominated by downdip slab sinking in the upper mantle and downdip sinking of the inclined folded slab pile. Now the poloidal return flow cell is long (~5000 km), scaling with the whole-mantle depth, dragging the

entire overriding plate, in particular its far backarc, towards the subduction zone hinge that resists retreat (Figs. 2c, 4b and 5e–h). This causes deviatoric compression and shortening in the overriding plate (Figs. 3a, b and 7a, b, d, 132.8 and 182.6 Myr), with maximum deviatoric compressive stresses of 20–80 MPa occurring within 1000 km from the trench. Note that the upper plate compression is not due to landward-directed (eastward) flow of the mantle below the overriding plate or a landward-directed gradient of upper plate motion, rather it is due to the gradient in trenchward (westward) motion within the upper plate, due to the trenchward mantle flow below the overriding plate (Figs. 2c and 5e–h).

Resistance to slab retreat is most significant in the centre of the wide subduction zone and decreases towards the lateral edges, because of the difficulty (longer lateral flow path) for sub-slab mantle material in the centre to move to the mantle wedge region in the centre[35, 43]. Therefore, compressive stresses are maximum in the centre (cf. Central Andes) and decrease towards the north and south (Fig. 7d), which is consistent with earlier work showing latitudinal variation in mantle drag, decreasing northward and southward from the Central Andes[36].

The geodynamic model demonstrates that cordilleran mountain building can be explained by subduction-induced poloidal

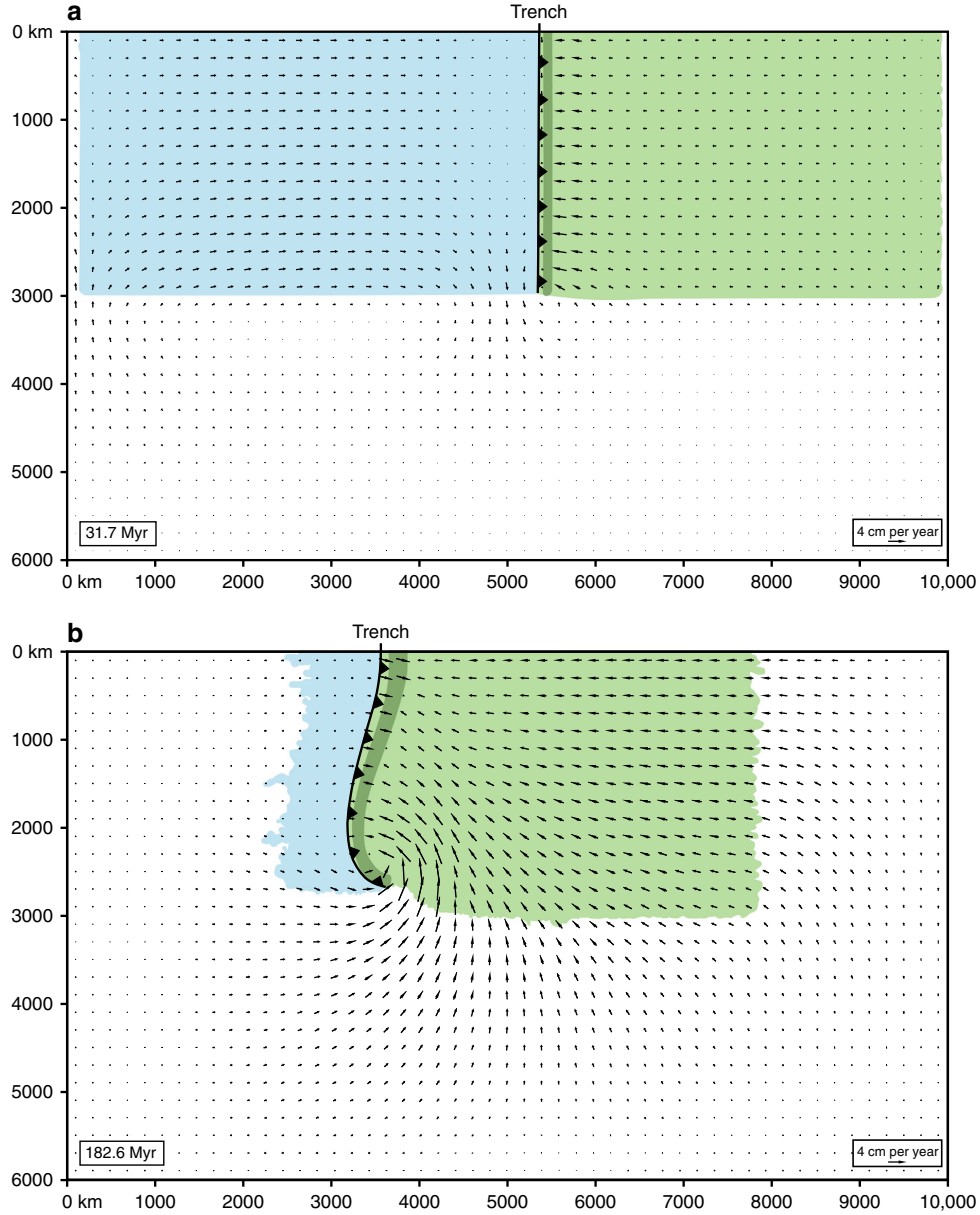

**Fig. 4** Model results showing the horizontal velocity field in the mantle at 200 km depth and the outlines of the tectonic plates at the surface. **a** At 31.7 Myr. **b** At 182.6 Myr. Also shown are the trench (black continuous line with black triangles) and the outline of the slab at 200 km depth (elongated dark green feature on the right-hand side of the trench running sub-parallel to the trench). Note that the subducting plate is in blue, while the overriding plate is in green

mantle flow in the deep mantle. It is thereby consistent with earlier conceptual models proposed for the Andes[36, 48] and Sevier-Laramide orogeny in North America[35], and is in line with earlier 2D numerical subduction models[37, 39]. Apart from confirming the importance of whole-mantle subduction and whole-mantle flow for cordilleran orogeny, as proposed earlier[35, 36, 48], the new model also demonstrates the crucial role of slab width and long timescales (~100–200 Myr) in Andean-style mountain building. The model further shows that the deep upwelling below and behind the far-field overriding plate is induced entirely by the deep mantle folded slab pile (Fig. 2c), which is in contrast to an earlier proposal in which upwelling in this location is thought to be primordial[36]. The geodynamic model also contrasts with the conceptual model of "slab pull–orogeny" involving the formation of moderately thick

crustal stacks during upper mantle subduction[48], as it shows significant backarc extension with some minor accretion (Figs. 2a, 3a, b and 8a) rather than orogeny.

**Crustal thickness evolution.** The crustal thickness decreases due to extension until ~67.5 Myr in the centre, after which it increases due to shortening to a maximum of 74 km at 182.6 Myr (Fig. 8a). Such a maximum crustal thickness can isostatically support a mountain topography of 6.2 km (±0.2 km) with respect to the undeformed continental lithosphere of South America. The maximum model crustal thickness is consistent with the maximum crustal thickness (70–75 km) in the Central Andes[20, 49] (Fig. 1e). The maximum crustal thickness achieved in the model evidently depends on the initial crustal thickness. A 30 km thick

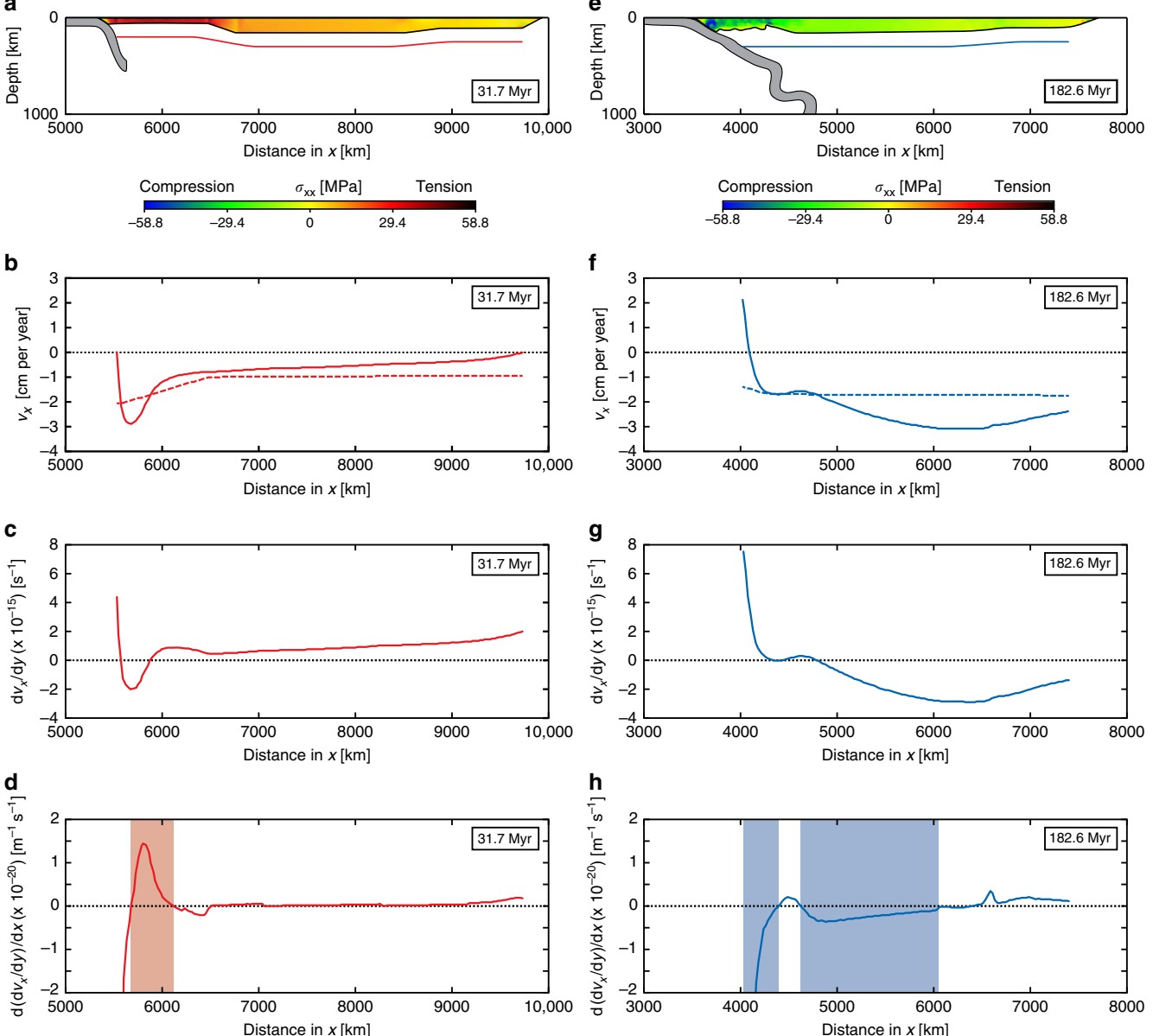

**Fig. 5** Model results showing the horizontal velocity and shear rate and shear rate gradient in the centre of the subduction zone below the base of the overriding plate at 31.7 and 182.6 Myr. **a**, **e** Location of line at ~150 km below the base of the overriding plate, with colour scheme in overriding plate indicating trench-normal horizontal deviatoric normal stress ($\sigma_{XX}$). **b**, **f** Horizontal trench-normal velocity in the mantle ($v_x$) at ~150 km below the base of the overriding plate (continuous line) and in the overriding plate (dashed line). Note that the direction towards the right (in the positive x-direction) is taken as positive. **c**, **g** Average horizontal trench-normal shear rate ($dv_x/dy$) in a ~150 km thick mantle section below the base of the overriding plate. Note that a basal drag in the direction towards the right (in the positive x-direction) is positive. **d**, **h** Trench-normal horizontal gradient of the average horizontal trench-normal shear rate ($d(dv_x/dy)/dx$) in a ~150 km thick mantle section below the base of the overriding plate. Note that a positive gradient induces trench-normal overriding plate extension, while a negative gradient induces trench-normal overriding plate shortening. The pale red zone in **d** for 31.7 Myr indicates a zone with a strong positive gradient, while the pale blue zones in **h** for 182.6 Myr indicate zones with a strong negative gradient

initial crust was chosen, which is reasonable for the Southern and Central Andes, considering the present-day far-field crustal thickness (30–35 km), but on the low side for the Northern Andes (35–40 km)[20]. However, what is important to note is that the driving mechanism of cordilleran mountain building is capable to both deform the overriding plate lithosphere and support the major buoyancy forces that come with 74 km thick continental crust. The model also reproduces the trench-parallel decrease in crustal thickness from the Central Andes northward and

southward to ~30–40 km in an advanced stage of subduction (Fig. 8b), which is consistent with observations (Fig. 1e).

**Evolution of trench curvature and trench migration.** Due to the trench-parallel variation in overriding plate deformation, with initially faster backarc extension near the edges followed by faster shortening in the centre, the active western margin of the overriding plate develops a progressive curvature in the centre that is convex towards the overriding plate (Fig. 3e). It is noteworthy

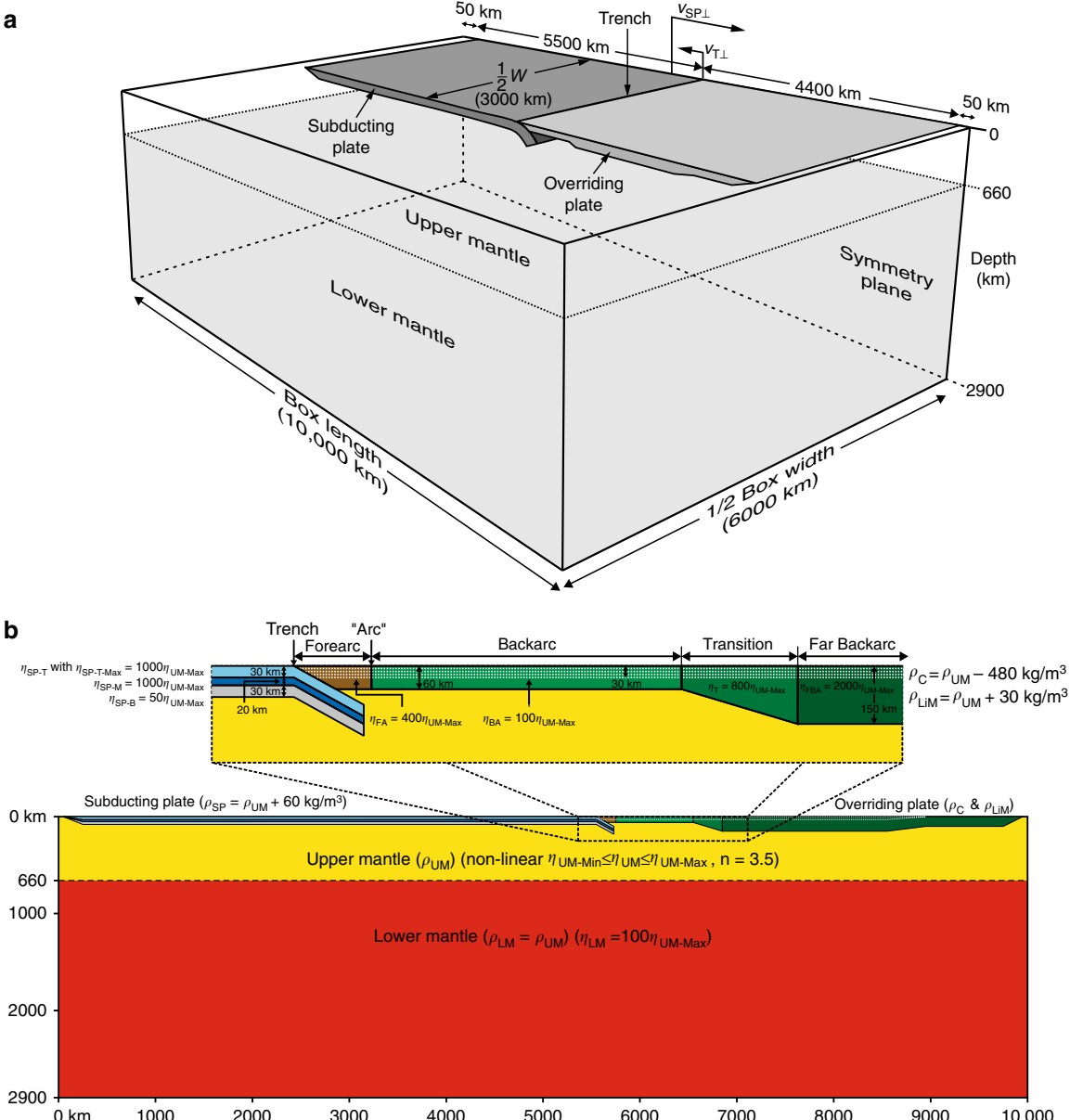

**Fig. 6** Numerical model set-up to investigate time-evolving subduction of a 6000 km wide subduction zone comparable to the South American subduction zone in a very large three-dimensional domain with a layered whole-mantle reservoir. **a** Three-dimensional perspective view. **b** Cross-sectional view through the centre of the subduction zone (symmetry plane in **a**). The model includes a layered negatively buoyant oceanic subducting plate with three layers with different viscosity (top, middle and bottom) and a layered continental overriding plate with two layers with different density (crustal layer and lithospheric mantle layer). Note that $v_{T\perp}$ = trench-normal trench velocity (oceanward retreat is positive), $v_{SP\perp}$ = trench-normal subducting plate velocity (trenchward is positive), $W$ = slab width, $\rho_{UM}$ = sub-lithospheric upper mantle density, $\rho_{LM}$ = lower mantle density, $\rho_{SP}$ = subducting plate density, $\rho_C$ = continental crustal density, and $\rho_{LiM}$ = lithospheric mantle density. See Methods section for more details

that the convex geometry develops during continuous trench retreat (Fig. 3a, e), which is slowest in the centre and increases towards the lateral slab edges with a maximum at ~500 km from the edges. Average trench retreat velocities as deduced from Fig. 3e give a minimum of 1.1 cm per year for the centre and a maximum of 1.3 cm per year near the edges over a 182.6 Myr period. Such rates are of the same order of magnitude as average South American trench retreat rates for the last ~200 Myr (~1.8 ± 0.3 cm per year in the centre and ~1.9–2.1 ± 0.3 cm per year near the edges) and are comparable to observed present-day rates of trench migration (−0.7 to 0.6 cm per year in the centre and 1.1 to 1.6 cm per year in the north and south)[43]. The trench velocities in nature also show the pattern of slower trench retreat rates in

the centre compared to the edges. Please note, however, that the dimensionalized velocity values of the numerical model directly depend on the choice of absolute viscosity for the mantle in nature (Methods section).

The evolution of trench curvature in the model (Fig. 3e) agrees with the development of the convex-shaped Bolivian orocline in the Central Andes[16, 18, 19], which has been developing since the Cretaceous as implied by paleomagnetic rotations[15]. Jurassic-Cretaceous rocks record a range of anticlockwise rotation values (~ 5–70°) north of the Arica bend and clockwise rotation values (0–50°) south of the bend[15]. The directions of these rotations (anticlockwise in the north and clockwise in the south) are consistent with the oroclinal rotations observed in the model.

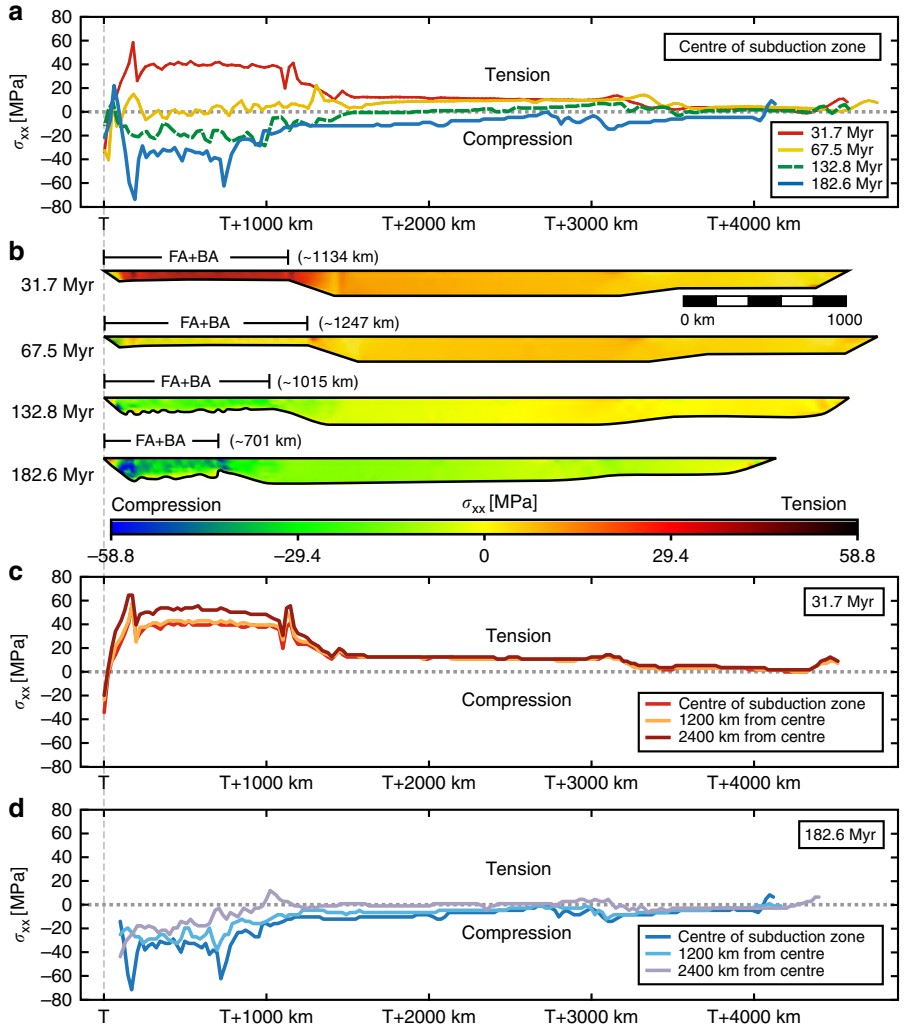

**Fig. 7** Model results showing the stresses in the overriding plate. **a** Trench-normal horizontal deviatoric normal stress ($\sigma_{XX}$) at the overriding plate surface in the centre of the subduction zone at four different times (31.7 Myr corresponding to the Middle Jurassic in nature, 67.5 Myr corresponding to the Early Cretaceous, 132.8 Myr corresponding to the latest Cretaceous, and 182.6 Myr corresponding to the Late Cenozoic). **b** Cross-sections showing overriding plate $\sigma_{XX}$ in the centre of the subduction zone at four different times. **c** $\sigma_{XX}$ at the surface at 31.7 Myr (corresponding to the Middle Jurassic) for three locations (in the centre of the subduction zone, at 1200 km from the centre and at 2400 km from the centre, i.e. close to the lateral slab edges). **d** $\sigma_{XX}$ at the surface for three locations at 182.6 Myr (corresponding to the Late Cenozoic). Note that T indicates the location of the trench. Also note that the highest tension in **c** occurs at 2400 km from the centre (i.e. close to the lateral slab edges), while the highest compression in **d** occurs in the centre of the wide subduction zone

However, maximum model rotations (±16°, Fig. 3e) are on the low side when compared to recorded measurements in nature. This might be partially due to the gentler curvature of the orocline in the model compared to the more acute curvature in nature, possibly resulting from the lack of brittle rheologies in the model, and/or partially due to local block rotations in nature. The model further produces concave (towards the overriding plate) subduction zone edges, which is in agreement with the northern and southern concave edges of the South American subduction zone (Fig. 1a), and can be ascribed to the toroidal mantle return flow around the lateral slab edges[43, 50, 51].

**Evolution of slab dip angle and arc migration.** The model demonstrates that on a 200 Myr timescale the upper-mantle to whole-mantle subduction evolution induces a progressive decrease in slab dip angle in the upper mantle, which is most pronounced in the centre and less pronounced away from the centre (Fig. 9a, b). In the centre the slab dip angle at 0–200 km depth decreases from 55° at 20 Myr to 26° at 190 Myr. The

trench-parallel variation is partly explained by the larger push force of the overriding plate and resistive force of the slab in the centre to retreat compared to the edge regions (because of the relatively immobile sub-slab mantle in the centre[43, 52]). Such higher forces are evidenced by the larger horizontal compressive stress in the centre compared to the sides (Fig. 7d), causing a greater reduction in slab dip angle in the centre. The decrease in slab dip angle is further explained by the larger suction force in the mantle wedge in the centre due to its greater distance from the lateral slab edges, which tends to lift the slab and reduce the slab dip angle. The slab folding at the 660 km discontinuity is responsible for periodic short-term (~20–30 Myr) changes in slab dip angle, most notably at 200–400 km depth (Fig. 9b), which are superposed on the long-term trend of slab dip angle decrease. The general pattern of increasing slab dip angle towards the lateral slab edges in the model is consistent with global observations showing steeper slab dip angles near such edges[53].

The long-term progressive change in slab dip angle in the model predicts a progressive eastward migration and widening of

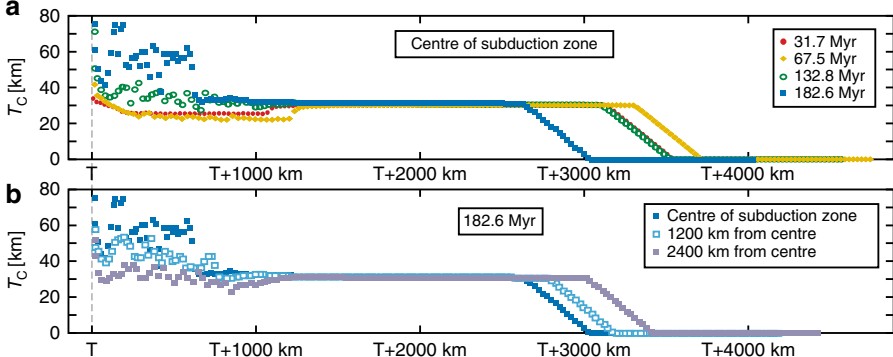

**Fig. 8** Numerical results showing the temporal evolution and lateral variation of the continental crustal thickness in the overriding plate. **a** Overriding plate crustal thickness ($T_C$) at four different times (31.7 Myr corresponding to the Middle Jurassic in nature, 67.5 Myr corresponding to the Early Cretaceous, 132.8 Myr corresponding to the latest Cretaceous, 182.6 Myr corresponding to the Late Cenozoic) in the centre of the subduction zone. **b** Overriding plate crustal thickness at three different sections along the subduction zone (centre of the wide subduction zone, 1200 km from the centre and 2400 km from the centre) in a late stage of the subduction evolution (182.6 Myr, corresponding to the Late Cenozoic). Note that the original continental crustal thickness (at model time = 0 Myr) is 30 km, and that changes in thickness are due to overriding plate shortening and/or extension

the magmatic arc in the centre of the subduction zone (Fig. 9c, centre). It can thereby provide an explanation for ~200 km of eastward (landward) migration and progressive widening of the magmatic arc in the Central Andes region since the Jurassic[7, 8, 11] (Fig. 1b, c). Notably, the geodynamic model can also explain the reduced/absent arc migration in the Southern Andes[54] and Northern Andes (Figs. 1a and 9c, 2400 km). For the Central Andes, the model predicts an eastward migration of 110–270 km and an increase in arc width (trench-normal) from 90 to 250 km over a 170 Myr period of subduction (Fig. 9c). At 2400 km from the centre, the eastward migration is only 20–60 km and the arc width increases only from 90 to 130 km.

**Continental crustal destruction or growth.** The eastward arc migration in the Central Andes has been interpreted as evidence for long-term subduction erosion of the South American plate[55, 56], and estimates of long-term global subduction erosion rates have been largely based on the South American case. Such estimates have been used to argue for destruction of considerable amounts of continental crustal material ($1.4 \text{ km}^3$ per year out of $3.2 \text{ km}^3$ per year total[56]) and little net continental crustal growth during the Phanerozoic[55, 56]. Indeed, the small separation between the Jurassic arc along the coast in southern Peru and northern Chile and the present-day trench, with a minimum of 74 km (±5 km), might suggest that significant subduction erosion must have occurred along the trench. However, the smallest separation between the trench and predicted magmatic arc in the centre of the subduction zone in the geodynamic model is 58 km (Fig. 9c, at a time of ~20 Myr). Thus, the subduction model can account for the location of the Jurassic arc with respect to the present-day trench in the Central Andes that formed in the early stage of subduction, without needing to invoke subduction erosion, although some subduction erosion in the region is still possible.

The subduction model can also explain 110–270 km of arc migration (Fig. 9c), which is 55–135% of the ~200 km of observed migration, implying much reduced or negligible subduction erosion rates since 200 Ma (i.e. 0–45% of the estimate, which is $0$–$0.6 \text{ km}^3$ per year). With global continental crustal addition estimates due to igneous activity of $2.7 \text{ km}^3$ per year[56], destruction due to sediment subduction and crustal detachment of $1.7 \text{ km}^3$ per year[56], and the new estimate of the global subduction erosion rate ($0$–$0.6 \text{ km}^3$ per year), a net continental crustal growth rate of $0.4$–$1.0 \text{ km}^3$ per year since 200 Ma is

predicted. This estimated growth rate is at odds with crustal models showing zero net growth[55–58] or a reducing continental crust[59], but is broadly consistent with models showing progressive continental crustal growth[60, 61], and fits well with a recent crustal growth rate estimate of $0.8 \text{ km}^3$ per year since the Late Archaean[61].

## Methods
**Numerical model.** The numerical simulation uses the code Underworld[6, 40–43], in which mantle flow is modelled in a three-dimensional Cartesian box by compositional buoyancy contrasts in an incompressible Boussinesq fluid at very low Reynolds number under isothermal conditions. The regional model has been designed to specifically investigate, on a long timescale (~200 Myr) and a large spatial scale (10,000 × 12,000 km laterally and whole-mantle depth, 2900 km) the driving force of deformation in South America, as well as the spatial and temporal variation in deformation, crustal thickness, slab geometry, slab dip angle and magmatic arc position. Distinct volumes are represented by sets of Lagrangian particles that are embedded within a standard Eulerian finite element mesh, which discretises the problem to solve the governing equations. For additional information on the numerical technique and the non-dimensional equations used, the reader is referred to earlier work[6, 40–43]. Velocities and stresses in the model are scaled following the scaling formulations presented in Schellart and Moresi[37].

The current model design builds on the numerical model set-up from Schellart and Moresi[37], who investigated narrow subduction zone models (slab width = 800 km) with an overriding plate in an upper mantle domain. In this work a model is presented that uses a very large box that is 10,000 km long and 2900 km deep (entire mantle depth) (Fig. 6). The width (trench-parallel extent) of the model box represents 12,000 km, but we model only half the width (6000 km) due to the geometrical symmetry along the centre plane and the very low Reynolds number. We include a sub-lithospheric upper mantle domain down to 660 km depth with non-linear stress-dependent viscosity with a stress exponent $n = 3.5$ (following Mackwell et al.[62]), maximum viscosity, $\eta_{\text{UM-Max}}$, and minimum viscosity, $\eta_{\text{UM-Min}}$, such that $\eta_{\text{UM-Min}} = 0.1\eta_{\text{UM-Max}}$. The variation in sub-lithospheric upper mantle viscosity was thus limited to one order of magnitude to facilitate reasonably rapid convergence in the numerical calculations. Note that the dimensionalized $\eta_{\text{UM-Min}} = 5 \times 10^{19}$ Pa s and $\eta_{\text{UM-Max}} = 5 \times 10^{20}$ Pa s. These values fall within the estimated sub-lithospheric upper mantle viscosity range ($10^{19}$–$10^{21}$ Pa s) in nature[63, 64], while $\eta_{\text{UM-Max}}$ is close to the estimated average sub-lithospheric upper mantle viscosity ($3$–$4 \times 10^{20}$ Pa s) as deduced from glacial isostatic studies and mantle convection studies[65, 66]. It should be noted, however, that uncertainty in viscosity values for the mantle in nature directly affect dimensionalized velocity values. Indeed, the scaling formulations dictate that, for example, a choice for the dimensionalized viscosity that is a factor of 10 larger results in a dimensionalized velocity that is a factor of 10 smaller.

The lower mantle reservoir is 2240 km thick and has a viscosity $\eta_{\text{LM}} = 100\eta_{\text{UM-Max}}$. As such, a minimum viscosity jump of a factor of $10^2$ was applied between the upper and lower mantle. This simple implementation of a viscosity step represents all the effects of the 660 km discontinuity (viscosity changes, density changes and thermodynamic reactions from mineral phase transitions) and captures the geodynamic essence of this discontinuity by reducing the lower mantle slab sinking velocity, as indeed implied by earlier numerical work on deep mantle mineral physics and phase transitions[67, 68]. In addition, more recent work has shown

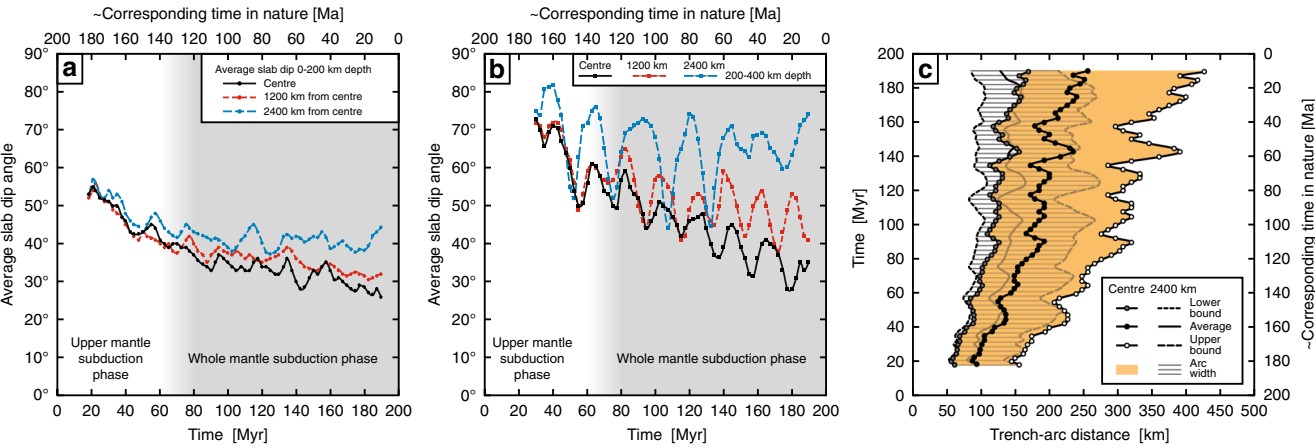

**Fig. 9** Numerical results showing the temporal evolution of the slab dip angle and magmatic arc migration. **a** Average slab dip angle at 0–200 km depth in three locations (centre of the wide subduction zone, 1200 km from the centre and 2400 km from the centre). **b** Average slab dip angle at 200–400 km depth in three locations (centre of the wide subduction zone, 1200 km from the centre and 2400 km from the centre). **c** Evolution of the magmatic arc position and width with respect to the trench in two locations (centre of the wide subduction zone and 2400 km from the centre) (for calculations see Methods)

that for studies in which a complete treatment of compositional layers and phase transitions is not implemented, the large-scale deformation of slabs is best approximated by a model with no phase transitions rather than including an incomplete approximation that over-predicts slab folding[69].

Subduction, plate motion, deformation and mantle flow are driven entirely by buoyancy forces and there are no imposed (non-zero) velocity or force boundary conditions. The bottom of the box represents the core-mantle boundary and is appropriately modelled with a free-slip boundary condition. The other boundaries of the box also have free-slip conditions. For the top surface this is a reasonable approximation considering that the viscoplastic rheology of the top layer of the subducting plate allowed for the subducting plate to decouple from the top surface at the trench, facilitating subduction[42]. Furthermore, the enormous size of the numerical modelling domain did not allow for a higher resolution near the top surface such that a free surface could be implemented. The numerical geodynamic model is a regional model that investigates the subduction process of the South American subduction zone in isolation (excluding far-field effects such as from other subduction zones or global mantle flow) to study if this subduction zone can explain some of the first-order characteristics of the Andean margin of South America. Therefore, the lateral boundary conditions (free-slip) and lateral size of the box were chosen such that their effects on the subduction process in the model were minimized. The effects of the free-slip lateral side walls on the model outcomes in the current work are indeed very small due to the very large lateral extent of the model box (10,000 km by 6000 km) and the large separation (many thousands of km) between the subducted slab and the lateral side walls. In earlier work the influence of lateral side walls on subduction dynamics using a free-slip boundary condition has been tested[35, 43]. These earlier works showed that the separation of the slab and the lateral side walls should be ≥0.5 times the width of the slab for these side walls to not significantly affect the evolution of subduction and mantle flow. Another requirement is that the separation distance should be larger than the thickness of the mantle reservoir (2900 km in the current model). These two conditions are both met in the numerical model (see Figs. 2 and 4).

The current numerical simulation follows earlier works in which a Cartesian representation is used for numerical subduction models with a large lateral domain[6, 35, 70, 71]. This Cartesian representation is a simplification of the spherical Earth, but earlier work implies that the subduction dynamics, subduction-induced mantle flow and the evolving slab and trench geometry are comparable in both Cartesian and spherical models[70]. Furthermore, the Underworld code used for the simulation (version 1.6) did not include a capability for depth-dependent compressibility, and it was therefore decided to use a Cartesian geometry.

We include an 80 km thick three-layer subducting plate simulating the Nazca/Farallon plate, assuming an average age of 40–50 Ma for the oceanic lithosphere and using a half-space cooling model to derive its thickness[72, 73]. The assumed age of the oceanic lithosphere is consistent with reconstructions of the oceanic lithosphere subducting along the South American subduction zone since 60 Ma, which imply that its age was mostly in the range 30–60 Ma[74]. The age of the oceanic lithosphere at the trench in the period 200–60 Ma is much less constrained and rather speculative, and due to this lack of constraints it is kept at a constant thickness. The subducting plate has a 30-km-thick viscoplastic top layer with a maximum viscosity ($\eta_{SP-T-Max} = 1000\eta_{UM-Max}$), a strong, 20-km-thick, Newtonian-viscous middle layer ($\eta_{SP-M} = 1000\eta_{UM-Max}$) and a weaker, 30-km-thick, Newtonian-viscous bottom layer ($\eta_{SP-B} = 50\eta_{UM-Max}$). The subducting plate is

laterally homogeneous and we ascribe it a density that is 60 kg m$^{-3}$ higher than that of the sub-lithospheric mantle. The surface part of the subducting plate is 6000 km wide (trench-parallel extent) and 5,500 km long (which includes a 200 km long tapered trailing edge) with an additional 206-km-long initial slab perturbation dipping at 29°.

The overriding plate is 6000 km wide and 4400 km long and has a thickness that varies in the direction normal to the trench. It contains a continental part (representing the South American continent) and an oceanic part (representing Atlantic oceanic lithosphere), and both domains have linear viscous rheologies. The frontal part of the continental region contains a 200 km long high-viscosity forearc ($\eta_{FA} = 400\eta_{UM-Max}$) and an 800 km low-viscosity backarc ($\eta_{BA} = 100\eta_{UM-Max}$) that are both 60 km thick, and a 300 km transition zone ($\eta_T = 800\eta_{UM-Max}$) with a thickness changing from 60 to 150 km. The 3100 km long far backarc ($\eta_{FBA} = 2000\eta_{UM-Max}$) has a 2100 km long, 150 km thick, (cratonic) continental part, and a 1000 km long oceanic region that is 100 km thick but includes a 200 km long tapered trailing edge. The geometries of the overriding plate forearc, backarc and cratonic part, and their relative strengths, are based on the work of Currie and Hyndmann[75] on the thermal structure of ten forearc-backarc systems at ocean-continent subduction zones in the Pacific domain. They found comparable geometrical set-ups at these zones, generally consisting of a cold, ~200–300 km long, forearc, a warm, ~600–1000 km long, backarc with a lithospheric thickness of ~60 km, followed in a number of cases by cold, strong and thick cratonic lithosphere in the far backarc implying a lithospheric thickness of 150 km or more. The continental part has a 30 km thick upper layer representative of the crust with an ascribed density that is 480 kg m$^{-3}$ lower than that of the sub-lithospheric mantle, while the continental lithospheric mantle and the oceanic lithosphere of the overriding plate have a density that is 30 kg m$^{-3}$ higher than that of the sub-lithospheric mantle.

Mesh resolution in the $10,000 \times 6000 \times 2900$ km numerical domain is 512 (length) × 160 (width) × 192 (depth) elements. A spatially adaptive mesh has been implemented such that a domain of 3000 km (length) × 290 km (depth) around the subduction zone has a maximum resolution with cells with spatial dimensions of 9.8 km (length) by 7.6 km (depth). The higher resolution in length and depth is essential for properly resolving the subduction zone interface[37]. Initial particle distribution is 20 particles per cell, resulting in an initial total of 314,572,800 particles. The model ran for 5010 timesteps on 496 processors involving a total of 194 restarts, used ~$1.5 \times 10^6$ CPU hours, and took more than 2 years to complete on Raijin, the Australian national supercomputer, starting in September 2014 and finishing in late 2016.

The absence of thermal gradients in the model, and with it the absence of warming of the slab, will not have a significant effect on the slab morphology, slab viscosity and slab-mantle density contrast in the upper mantle due to the relatively rapid rate of subduction (up to 7.5 cm per year) and the slow rate of slab warming through conduction. Warming of the slab in the lower mantle would be more significant, and would likely produce a weaker slab in the lower mantle, which would thereby likely result in stronger slab folding producing tighter slab fold structures in the lower mantle. However, the density contrast of the lower mantle folded slab pile, which sinks as a whole and includes the entrained mantle material enclosed within the folds, would not be significantly affected, as the warming of the folded slab segment coincides with the cooling of the entrained ambient mantle in the folds. As such, the thermal buoyancy contrast of the entire folded slab pile would not diminish and disappear on a timescale of 100–200 Myr, and

so the driving mechanism of the lower mantle slab would not be significantly affected.

In order for a subduction model to be a valid first-order approximation of subduction in nature, it needs to show plate-like behaviour of the plates, single-sided subduction and strain localization at and close to the plate boundary. The current model meets these criteria, showing progressive single-sided subduction, plate-like behaviour of the two plates, which mostly retain their rectangular shape, and localization of deformation near the subduction zone interface.

**Calculations of arc width and arc position**. In the numerical model the calculations of the predicted magmatic arc width and the predicted magmatic arc position with respect to the trench were made using results from the global observational database of Syracuse and Abers[76]. For the lower bound, average and upper bound of the magmatic arc position, the top of the slab was assumed to be at 83, 125, and 208 km depth, respectively, using the values of Syracuse and Abers[76] in which hypocentre errors were taken into account. Please note that there is no implementation in the numerical model to simulate magma genesis.

**Jurassic magmatic arc position**. The position of the Jurassic magmatic arc in South America as plotted in Fig. 1a is based on the following sources: Hervé et al.[77], Noble et al.[78], Ramos and Folguera[54], Oliveros et al.[79], Sempere et al.[80], Suárez and Márquez[81], and Zapata et al.[82]. Note that the location for the present-day magmatic arc as plotted in Fig. 1a is based on Syracuse and Abers[76].

**Subduction erosion and continental crustal growth rate**. The new estimates of the global subduction erosion rate and continental crustal growth rate have made use of relatively recent data presented in Scholl and von Huene[56]. A similar estimate of the continental crustal growth rate is obtained, however, when using older data presented in Reymer and Schubert[60] and von Huene and Scholl[55]. The global subduction erosion rate by von Huene and Scholl[55] is estimated at 0.6–1.1 km$^3$ per year, which is 46–61% of their total rate of continental crustal destruction. As mentioned in the main text, the numerical subduction model can explain 110–270 km of arc migration (Fig. 9c), which is 55–135% of the ~200 km of observed migration, implying much reduced or negligible tectonic erosion since 200 Ma (i.e., 0–45% of estimate). The numerical subduction model implies lower rates of long-term subduction erosion of 0–0.5 km$^3$ per year. Using global continental crustal addition rates due to igneous activity from Reymer and Schubert[60] and von Huene and Scholl[55] (estimated at 1.65 km$^3$ per year for the last 600 Myr), sediment subduction at 0.7 km$^3$ per year[55], and the new estimate of the global subduction erosion rate (0–0.5 km$^3$ per year), a net continental crustal growth rate of 0.45–0.95 km$^3$ per year since 200 Ma is predicted using the older data. This estimate is comparable to the estimate presented in the main text (0.4–1.0 km$^3$ per year) using more recent data, is broadly consistent with models showing progressive crustal growth[60, 61], and also fits well with a recent crustal growth rate estimate since the Late Archaean[61] of 0.8 km$^3$ per year.

**Code availability**. The Underworld code is an open-source numerical particle-in-cell finite element code that has been specifically designed to simulate large-scale geodynamic processes. The Underworld code is available at http://www.underworldcode.org.

**Data availability**. All the data generated by the numerical model that are necessary to evaluate this work are included in this published article. All the data and information that are required to reproduce the numerical model results are presented in this published article, and the Underworld code that can be used to reproduce the model is openly accessible.

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

## Acknowledgements

Discussions with Zhihao Chen, Pieter Vroon, Gordon Lister, Joao Duarte, Vincent Strak, and Louis Moresi on subduction dynamics, Andean geology, mountain building, continental crustal growth and computational geodynamics are greatly appreciated. I would like to thank Louis Moresi, Mirko Velic, Julian Giordani, John Mansour, and Owen Kaluza for technical support with, and continuous development of, the Underworld code. I would also like to thank the reviewers Nadine McQuarrie, Laurent Husson, and Margarete Jadamec for their helpful comments. This work has been funded by a Vici Fellowship (016.VICI.170.110) from the Dutch National Science Foundation (NWO), and has been supported by computational resources from the NCI National Facility in Australia through the National Computational Merit Allocation Scheme (project ei8).

## Additional information

**Competing interests:** The author declares no competing financial interests.

