## [Peer Review File · Nature Communications]

Reviewers' comments:

Reviewer #1 (Remarks to the Author):

Review Summary:

This paper presents a large-scale time-dependent three-dimensional model of subduction to test hypotheses for the formation of the Andes, magmatic arc migration, and variations in crustal thickness along the Andean margin in South America. The origin of the Andes and evolution of South America continues to be a hotly debated topic, despite the number of previous studies. As numerical modeling capabilities are advanced, more sophisticated hypothesis can be tested. This paper presents a significant contribution to the origin of the Andes. Most models of this scale do not examine thickness variations in the crust of the overriding plate over time, but rather examine dynamic topography as a proxy for mountain building. The approach here is thus a large step forward and raises provocative questions as to how to incorporate and interpret those effects into a large-scale 3D model. This paper will be widely read and received, and will have an impact across disciplines within geoscience, from the geodynamic modeling community, to field geology and tectonics, to igneous petrology, as well as the general community who studies the Andes. I recommend the paper for publication in Nature Communications. The detailed comments below include both conceptual comments on the model assumptions, additional relevant literature to address, as well as editing comments.

Main Text:

Line 27 – 'whole mantle' might be perceived as all of the Earth's mantle, i.e., a global model. However, this is a regional model. What is meant here, is that the model depth spans the depth extent of the mantle. Please clarify.

Lines 38-40 – Mountain building has also been attributed to coupling between the upper plate with an underlying flat slab and/or lithospheric discontinuities, e.g., Jadamec et al., 2013 EPSL; Haynie and Jadamec Tectonics 2017.

Lines 47-52 – Split sentence.

Line 97 – The connection in time might be confusing. I understand that the difference is because the model time starts at 0. Perhaps better clarify for the non-modeler reader.

Line 1113-1118 – Please comment on this large-scale mantle flow driving mechanism in

the context of the recent results of Faccenna et al., 2017.

Lines 135-142 – Split sentence.

2

Lines 144-154 – This would be a good place to help the reader interpret the results in Figure 4. That is, that, the results as plotted indicate the curvature of the trench, with a landward apex in the centerline, is not due to a phase of trench advance. It is due instead to a reduced relative magnitude of trench retreat in the centerline of the slab, compared to the distal segments of the trench near the lateral slab edges. Also along these lines, either here or elsewhere, it would be useful to clarify that the compression in the upper plate is not due to landward-directed flow of the upper plate or mantle (i.e., not toward the right, in Fig. 3), rather it is due to the gradient in trenchward motion within the upper plate (i.e., gradient in leftward upper plate motion in Fig. 3), due to the trenchward mantle flow.

Including an upper plate, and one with a relatively realistic lateral variations in lithospheric thickness is a significant contribution of this paper. Can the author comment on the overall significance of including an upper plate in these models resulting trench curvature and slab dip (e.g., Holt et al., 2016; Sharples et al., 2014).

Please comment distinguishing the results here from other models examining slab geometry in South America (Martinod et al., 2005; Hu et al., 2016; and particularly, Faccenna et al., 2017).

In addition, please comment on the expected effects of phase transitions (e.g., Bina et al., 2001; Arrendondo and Billen, 2016) on the slab dynamics, trench motion, and folding character of slab and trench motion, which in turn may also affect upper plate deformation in the model.

Lines 156-171 – It may be worth pointing out also that the steeper slab dip at the lateral slab edges is consistent with the overall observed pattern of steeper slab dips near lateral slab edges on earth (e.g., Lallemand et al., 2005).

Line 184-201 (and in methods)– This is somewhat of a new approach, and therefore additional text would help here or in the Methods. Specifically state either here or in the methods that Figure 8 shows the predicted change in thickness due to crustal shortening or extension, i.e., the vertical measurement of the bottom and top of crustal layers at the specified locations in the model. The discussion then, is about placing these numbers in

context. Along these lines, there is no magma-genesis in the models. This should be specified in the methods. So, there it is understood that there is no contribution from the mantle in terms of accreting material to the base of the upper plate or within the plate through pluton solidification or a volcanic contribution. This will make it more clear that the results give bounds on tectonic contribution. Can the author comment on expected magma-genetic contribution from melting in the mantle & the addition of material to the upper plate from below?

Methods:

Line 208 – Add Underworld reference.

3

Line 209-210 – Is the Cartesian representation appropriate for a large swath of spherical earth. Please provide estimate why this assumption is appropriate.

Line 210 – Give number as well as 'whole mantle depth' (2900 km).

214 – The mesh doesn't solve the problem. I know the author knows this, the sentence is just awkward as written. The Eulerian part of the problem is solved on the mesh?

226 – Awkward at end of sentence. Try this "..., a maximum viscosity, $\eta_{UM(Max)}$, and a minimum viscosity, $\eta_{UM(Min)}$

, such that $\eta_{UM(Min)} = 0.1\eta_{UM(Max)}$

."

Thus, there is only a factor of 10 difference allowed between the maximum and minimum viscosity in the upper mantle. So, there is no more than 1 order of magnitude variation in the upper mantle viscosity. It would be useful to mention here or in the methods the scaled values, i.e., does this represent 10^{19} to 10^{20} Pa s or 10^{20} to 10^{21} Pa s? How might larger viscosity variations be expected to affect the result (e.g., Billen and Hirth, 2006; Jadamec, 2016).

Line 232- What is the relative thickness of the three sublayers? Okay, I see this is in the model set-up figure (30 km, 30 km, and 20 km). Also specify that here?

Line 233-234 – Specify, assuming correlation between plate thickness and age from $\frac{1}{2}$ space cooling model. Add reference, i.e., Turcotte and Schubert (2014) would suffice.

How did the seafloor age vary for this region through time from plate reconstructions for the Nazca and Farallon plates? How would you expect this assumption affect the result,

for example the younger ridge subduction (Sdrolias and Muller, 2008)?

Also, there is no temperature evolution in the model. Please specify this in the methods to help the reader understand the model limitations and implications.

How would you expect the slab morphology (width), thermal structure (density), and viscosity (temperature dependent weakening) to change over time due to diffusion and advection of heat in the mantle and warming of the slab?

Similarly, the model simulates subduction for a sustained amount of time ~ 200 My. How might the thermal evolution be expected to affect the continental growth and possibly continental delamination or drips of mantle lithosphere from the overriding plate over time?

Lines 235-236 and subsequent parts on viscosity – It would be useful somewhere to dimensionalize the viscosity. If η_{UM} is 1020 Pa s, then the viscosity of the upper and middle layer of the subducting plate would be 1023 Pa s, and the viscosity of the forearc and backarc would be on the order of 1022 Pa s. Alternatively if the η_{UM} was 1019 Pa s, the oceanic and continental plate viscosities would be on the order of 1022 Pa s and 1021 Pa s, respectively. It would be useful to compare estimates for upper plate viscosity based on 4

observations (e.g., England and Molnar, 1997) and theory (Jadamec et al., 2007), albeit these references are for the Tibetan plateau.

Line 251- “upper layer representative of the crust” rather than ‘crustal layer’. Or is there elasticity to this layer? What equation governs its rheology?

Line 276 - Isn't this 72 km, 105 km, and 173 km, respectively (Syracuse and Abers, 2006). Please clarify.

Lines 262-263 – Specify the total number CPU hours for the model as well (e.g., 496 CPUs * X hours).

Lines 278-281 – Why are only references for the Jurassic arc given here? What about constraints for the younger arc?

Line 283 – Please clarify based on comments for Lines 184-201.

Figures:

Figure 1 – Is the horizontal axis for c same as in a? If so, add ticks. If not, why not? Is the trench normal distance for Peru through time not constrained?

Figure 2b – If I understood the methods correctly, then in the upper mantle (yellow layer in the larger domain image) specify: $\eta_{UM}(\text{Min}) < \eta_{UM} < \eta_{UM}(\text{Max})$

Figure 3 – Specify v' is non-dimensional velocity.

Figure 4a – This is a conceptual/understanding of results comment. I added it also in the comments where Figure 4 is referred to in the text. The blue line in Figure 4a is positive the whole time (always retreating). In Figure 4e there is a trench curvature change, such that the final curvature at time 182.6 has a landward apex in the center. This occurs even though the trench along the centerline retreats the entire time, in absolute velocity. Thus the landward directed trench apex is a result of relatively less retreat along the centerline than at the edges (not due to trench advance in the centerline region). It might be useful to point this out, because the curvature might be thought to be due to trench advance along the centerline.

Figure 4a – What is the significance of a negative subducting plate velocity (black line) in the very early phase of subduction (first 18 million years of model)? Is that an numerical artifact or is there something pulling the surface part of the subducting plate backwards (away from the trench) in the early phase?

Figure 4a,b,c – The connection and between the upper plate deformation measurements in (a) and (b) are not clear. The signal changes from positive to negative at about 70 Ma

5 in (a), but not until about 140 Ma in (b). I know these represent different quantities, but is not clear intuitively how the values in b,c relate to the deformation rates in 4a.

Figure 5 – This is a nice way to show the relative position of the trench and slab in a mantle depth slice.

Figure 6 – This is somewhat of a cosmetic comment, but might make this set of compilation figures easier to understand at first glance. Use two of the σ_{xx} scale bars – one under the upper right in left column and the other in upper right of right column. It will make it easier to see at a glance that the entire left column is results on one timestep and the right column is the same set of results but at a later timestep.

Figure 6b – Specify which direction is positive (toward or away from trench).

Figure 7c – It is very hard to tell the two dark red lines apart (the centerline and 2400 km away from centerline lines).

Figure 8 – This is a very significant contribution. Very few large-scale 3D numerical models of subduction quantitatively predict crustal thickness variations in upper plate over time. See questions in earlier part of review to clarify how this plot relates to the discussion.

Figure 9c – It is really hard to distinguish the two domains.

References referred to in review:

Arredondo, K.M. and Billen, M.I., 2016. The effects of phase transitions and compositional layering in two-dimensional kinematic models of subduction. *Journal of Geodynamics*, 100, pp.159-174.

Billen, M.I. and Hirth, G., 2007. Rheologic controls on slab dynamics. *Geochemistry, Geophysics, Geosystems*, 8(8).

Bina, C.R., Stein, S., Marton, F.C. and Van Ark, E.M., 2001. Implications of slab mineralogy for subduction dynamics. *Physics of the Earth and Planetary Interiors*, 127(1), pp.51-66.

England, P. and Molnar, P., 1997. Active deformation of Asia: from kinematics to dynamics. *Science*, 278(5338), pp.647-650.

Faccenna, C., Oncken, O., Holt, A.F. and Becker, T.W., 2017. Initiation of the Andean orogeny by lower mantle subduction. *Earth and Planetary Science Letters*, 463, pp.189-201.

6

Haynie, K. L., and M. A. Jadamec (2017), Tectonic drivers of the Wrangell block: Insights on fore-arc sliver processes from 3-D geodynamic models of Alaska, *Tectonics*, 36.

Holt, A.F., Buffett, B.A. and Becker, T.W., 2015. Overriding plate thickness control on subducting plate curvature. *Geophysical Research Letters*, 42(10), pp.3802-3810.

Hu, J., Liu, L., Hermsillo, A. and Zhou, Q., 2016. Simulation of late Cenozoic South American flat-slab subduction using geodynamic models with data assimilation. *Earth and Planetary Science Letters*, 438, pp.1-13.

Jadamec, M.A. and Billen, M.I., 2010. Reconciling surface plate motions with rapid three-dimensional mantle flow around a slab edge. *Nature*, 465 (7296), p.338.

Jadamec, M.A., Billen, M.I. and Roeske, S.M., 2013. Three-dimensional numerical

models of flat slab subduction and the Denali fault driving deformation in south-central Alaska. *Earth and Planetary Science Letters*, 376, pp.29-42.

Jadamec, M.A., Turcotte, D.L. and Howell, P., 2007. Analytic models for orogenic collapse. *Tectonophysics*, 435(1), pp.1-12.

Lallemand, S., Heuret, A. and Boutelier, D., 2005. On the relationships between slab dip, back-arc stress, upper plate absolute motion, and crustal nature in subduction zones. *Geochemistry, Geophysics, Geosystems*, 6(9).

Martinod, J., Funicello, F., Faccenna, C., Labanieh, S. and Regard, V., 2005. Dynamical effects of subducting ridges: insights from 3-D laboratory models. *Geophysical Journal International*, 163(3), pp.1137-1150.

Sdrolias, M. and Müller, R.D., 2006. Controls on back-arc basin formation. *Geochemistry, Geophysics, Geosystems*, 7(4).

Sharples, W., Jadamec, M.A., Moresi, L.N. and Capitanio, F.A., 2014. Overriding plate controls on subduction evolution. *Journal of Geophysical Research: Solid Earth*, 119(8), pp.6684-6704.

Turcotte, D.L. and Schubert, G., 2014. *Geodynamics*. Cambridge University Press.

Reviewer #2 (Remarks to the Author):

Schellart presents an intriguing model linking the early extension (~ Jurassic to early Cretaceous in age) and the subsequent switch to compression to early upper mantle subduction (backarc extension), followed by whole mantle subduction to produce the Andean orogeny. As stated in the abstract the strength in the model is that it does reproduce the maximum shortening and crustal thickness observed in the Central Andes and their progressive northward and southward decrease. The downside of the model that is not acknowledged anywhere in the manuscript is that it hugely (hugely) over predicts the magnitude of extension both the limited extension in the central Andes and the more prevalent extension in the southern and northern Andes. And thus also over predicts the amount of shortening in the southern Andes (by an order of magnitude) While I do not think that this is a reason to not publish the paper or the model (in this journal), I think it is disingenuous and limiting to not acknowledge that problems exist and not discuss why (why the over prediction, mis-prediction). Thus the short version of this review is that I think the manuscript should be published in *Nature Communications* but I also think that the author needs to acknowledge the problems and thus discuss potential solutions. Most of what I have added below addresses matching the model to data.

Magnitude ---To give an idea about the magnitude, the Basin and Range in the western US has undergone ~250 km of extension. This extension has produced metamorphic core complexes that rapidly exhume basement, produced multiple (10-20) normal fault bounded basins that have accumulated 3-6 km thick Tertiary basin fill during this extensional period. (~ 35-40 Million years). 250 km is the low amounts of extension that is modeled in the Central Andes where there are limited structures and sedimentary rocks that would support extension. (This is displayed in Figure 1 on the paper). Moving from south to north there are extensional basins as far north as 20° S, and a support for extension in rocks preserved along the coast starting again at 15° S. There is

really nothing, particularly in the folded and faulted rocks that comprise the central Andes between 20 and 15, that support anything but really limited (a few kms) of Jurassic – early Cretaceous extension in that region. While there is a lot to support extension of that age farther south (such as Salta Basin, central Chilean basin, Neuquen Basin) that any of these basins accommodated the minimum 250 km and the proposed 400 extension is a bit far fetched. The amount of continental extension that preceded the opening of the Atlantic Ocean ranges from 60 to 600 km, with many segments falling into the 200-400 km range of extension suggested here (point is with that much extension an ocean basin opens). Extension creates basins and preserves sedimentary rocks and those rocks get caught up in the subsequent compressional fold thrust belt. Thus it is not possible to just dismiss the lack of measured extension values with arm waving that the basins were inverted, eroded, “hidden” by later compression etc. The rocks that accumulated from that period would have been incorporated into the system and there has not been that much erosion in the Andes. There large thermochronometer data sets that allow us to track the thermal history of the rocks that are there and the surface rocks have not been buried by large sedimentary basins that have been erosionally removed. The point of this paragraph is to argue (strongly) that the extensional magnitudes presented in the model are wrong. Why? And what does that imply about our understanding of the system.

The other problem with having too much extension is that then there is too much compression need absorb the excess crust and come up with modern thickness and shortening values. However particularly north and south of the central Andes these shortening magnitudes are orders of magnitudes higher than documented. Again for example the Neuquen Basin in Argentina (Horton et al., 2016 cited in the paper) gives shortening estimates of 15 -30 km (documented in the inverted basins on the eastern side). The model argues for 300 km of compression. The 500 km of compression proposed for the central Andes is in the ballpark (particularly when accounting for the hard to document but certainly present shortening in the western cordillera that *is* Cretaceous in age and mostly covered by modern volcanics)

I think it would be insightful to all thinking about these processes to demonstrate not only the parts that agree but to highlight the misfits and the misfit magnitude.

Rotation --- Another interesting implication. How does it match with measured values? This can be done (easily enough), emphasize the magnitude of whole lithosphere rotation produced in the model, its age, and compare it to measured rotations of the CARP. Versus simply saying CARP argues for counter clockwise rotations north of the bend and clockwise south and look the model produces that too. Without doubt, the measured rotations are subject to local structural influences, have large uncertainties, and were sampled from lithologies older than Late Cretaceous (thus may record rotations unrelated to Andean deformation). But, you can also focus on paleomagnetic data from synorogenic sediments located in the Subandean region (compiled by Roperch et al., 2006; republished in Eichelberger and McQuarrie 2015) and highlight the magnitude that would be occurring from 15 Ma to present (south limb: 12°; north limb: -6°).

Crustal Thickness --- Any argument of amount of shortening and resulting crustal thickness is based on the assumption of original crustal thickness. The Kley and Monaldi curve shown in figure 1d assumes an initial (pre-shortening) thickness of 40 km. Decreasing the initial crustal thickness will of course require more shortening to account for the modern crustal thickness and have implications for amount of material available for parts of the Andes with less shortening and implications for recycling of crust and lower crustal flow. Saying the original crustal thickness of the Andes is 40 km is quite generous and was used by Kley and Monaldi back when shortening estimates were significantly lower than what was needed to explain modern thicknesses. We used it as a way of including the thickness of tertiary foreland basin rocks (5-7 km) on top of an original 35-33 km thick crust (Eichelberger et al., 2015, EPSL). But also show a 35 km version and a variable thickness version with eastern Cordillera thicknesses as low as 30 km (10-15 km sedimentary rock thickness and 15-20 km of basement). Again because of the preserved 12 km thick Paleozoic sedimentary rock record across the fold and thrust belt where the shortening estimates are constructed—it is hard to argue for 250 km of extension. This is $>1/4$ of the original length of the restored cross sections (~800 km) and hard to do without leaving a trace.

As a clarification any balanced cross section takes into account shortening east of the volcanic arc. For Bolivia the highest estimates are 350 km of shortening, with that increasing to about 400 km of shortening taking into account map view shortening (Eichelberger and McQuarrie 2015, GSAB). The shortening estimates in McQuarrie et al, 2005 that are 530-580 km of shortening state that balanced cross sections account for 330 km but also postulate ~200 km of shortening in the western cordillera that predates shortening to the east to account for preserved foreland basins. We do know that there is some magnitude of Cretaceous crustal shortening there, However we do not know the extent. The amount necessary to produce the basins preserved in Bolivia is 150-200 km. If the system switches to compression at 80 Ma, there is 30 Ma to deform the western Cordillera before deformation is solidly in the eastern cordillera (Rak et al, in revision) so the rates are ~ 5-7 mm/yr (which work). This additional amount can also get added to the systems in the south but that only increases the permissible shortening there from 50 km to 250 km—still short from in the 300 in the model but getting closer.

Of course the larger mismatch problem is farther south (and north) where the shortening estimates are much, much less than that needed to produce the measured crustal thickness. (as highlighted in Kley and Monaldi) the shortening estimates are more refined now but problem north and south of the Bolivian Andes remain. (e.g. Eichelberger et al., 2015) it is always possible for some model to predict the amount of shortening necessary to produce the right thickness. The problem is –did that amount of shortening actually occur.

Evolution of dip angle and subduction erosion--- So for me, the argument for subduction erosion has always been the occurrence of the Jurassic arc rocks *at* the modern coast and within 25-75 km of the modern trench. So instead of simply arguing slab angle can solve the problem completely (it really can not viably explain the arc rocks (plutons) 25 km from the modern trench) what is the minimum amount of subduction erosion needed to account for that (taking into account model dip angles)

Parts of the model that I really like: I think it is intriguing that even with the over prediction on the south (and possibly over prediction in the north) the difference in predicted shortening predicted in the central part of the model and that predicted to the south is ~ 300 km – similar to the measured differences (350-400 in the central Andean plateau and ~ 500-100 to the south). I am intrigued by the resulting predicted variation in shortening rates (and the cyclicity of them) cyclicity in Andean shortening was proposed by DeCelles et al., 2009 (Nature Geo) albeit the driver they propose is different. We also found cyclicity in shortening rates in the central Andes (Rak et al, in revision, tectonics). The timing of when extension and shortening happens is really good (even though magnitude could be way off).

Clearly this is a long review due to an obvious interest in the paper and its implications. It definitely caught my interest and I enjoyed reading the paper and thinking about these problems. My opinion is the paper will be stronger, and more useful if the author acknowledges both the fits and the mis-fits and presents ideas for what are driving the misfits in the model (not just attribute it to uncertainty in the data—there definitely is uncertainty but that uncertainty will not get you 250-400 km of extension!)

Small points:

Line 27 I think “demonstrating” is too strong.

Figure 1d I think there is a way to make this more interesting and more useful. I would also cite Schepers et al., 2017 for this because although their curve is not that far off of Arriagada, it is based on actually shortening data (Eichelberger et al., and updated shortening references north and south) that are not in Arriagada (theirs is model based). I thought that Schepers actually published the data for this in a supplementary info file (but can not find it). You can also put a dashed line above the Schepers line accounting for the ~ 200 km of shortening that is documented but super hard to quantify in the western cordillera (verses just a point). (citing McQuarrie et al., 2005 and Rak et al., 2017) and postulate this exists all along the coast (maybe??). What I think would be interesting and helpful is to plot your model predicted shortening over the actual shortening estimate lines (either in this figure or another figure)

As stated previously it is important to state that the line based on crustal thickness is assuming a 40 km thick initial crust. (This of course can get replotted assuming any initial crustal thickness).

Figure 2. It took me zooming in on this figure to find out the initial crustal thickness was 30 km (as I state above I think that is the thinnest one could go for initial Andes). The modern crustal thickness of the foreland basin is 40 km and has 3-6 km of foreland basin sediments on it and about 7 km of Paleozoic rocks. Thus 30 km basement is bare minimum. One could assume the basement thins and sediments increase to the west and that it stays 30 km. Particularly if you are discussing crustal thickness evolution you should state the initial crustal thickness (I know it is on figure 2 and 8 now but I felt I had to dig for it) and how it varies spatially (figure 8 a is great for the center but I think that plot at 1200 km out and 2400 km out would be insightful as to potential problems).

best regards

Nadine McQuarrie

Reviewer #3 (Remarks to the Author):

Review of Nature Communications manuscript NCOMMS-17-13241, by Wouter Schellart, entitled « Andean mountain building and magmatic arc migration driven by subduction-induced whole mantle flow ».

Laurent Husson, July 4th, 2017

Wouter Schellart presents a numerical model that aims at explaining the principal features of the Nazca - South America convergence: slab dynamics, mountain building, trench geometry, magmatism etc... all diagnostic processes that have been examined previously in numerous studies, but perhaps not in such a comprehensive manner. The main outcome of the model is to reproduce most of these feature using a very simple model (yet, simple doesn't necessarily mean too simple), that nevertheless is itself an impressive -and expensive- numerical performance (2 years computation time on 496 processors!). This work is remarkable in that sense.

From a scientific perspective, this model follows a very long series of cartesian subduction models, that present the advantage of delivering clear and easily understandable messages, but the drawback of not being fully consistent; the Earth is not cartesian, it is the thermal convection of the Earth that drives plates, and plates constantly interact together. Both methods are insightful, but it may be critical in many cases, as it is for the Andes (as proposed as early as 1998 by Lithgow-Bertelloni et al., Science).

Something that limits the attraction power of this paper is the fact that these results build on very many earlier studies, including some by the author himself, and is in that sense not so innovative. One merit is perhaps to provide a unifying view of such prior studies, but in many cases the conceptualization is not new. Examples are:

- An important result is the transition between upper mantle driven convergence and whole mantle driven convergence: the shallow slab excites a small size convection cell that promotes slab rollback, while the deep slab conversely drives major convection cells that promote the development of large orogens.

This is the core of a paper by Faccenna et al, 2013 (Tectonics, not cited), although they do not prove it experimentally.

- Another important is the fact that the upper plate could be dragged westward by the return flow of the subjecting slab. This is already explored and quantified by a force balance, by Wdowinski and O'Connell, 1991 (not cited).

- Trench curvature associated to the westward migration of the the slab is proposed by Russo and Silver to (1998?), followed later by Schellart et al (2007).

- Many of the current results also belong to Capitanio, or Uyeda and Kanamori (1982), or Martinod

et al., to cite a few.

- Presentation of earlier models (L. 56-58) is partial and doesn't give justice to the successfulness of earlier attempts. Of course, this is not the place for a review, but many other models exist, some explaining most features just as well. I can for instance refer, with some embarrassment, to Husson et al., 2012 (EPSL), which is in the same lines as the current paper, with a noticeable difference that upwellings are thought to be primordial, unlike in the current study, which is interesting.

- Model limitation:

* perhaps one strong limitation are the boundary conditions around the South American plate. At the beginning of the experiment, the plate touches the right end of the box, where BC are free slip (and hence no horizontal velocity is permitted). It takes some work to make it move to the center of the model. This is expressed as a resistance to displacement, and thus, adds up an extensional force in the upper plate. This force gradually disappears as the plate moves to the center of the box, and this occurs when compression occurs too. Could this be quantified?

* free-slip at all boundaries (L. 229). I assume this is also true at the surface. I'm a little out of date on these aspects, but I feel that many recent papers have shown the force balance to be different with free surface and free slip, hence different crustal growth scenario. How important is that?

* The setup / initial conditions are very important. Choice is made to have a 800 km wide low viscosity back-arc, 60 km thick only, with no justification. This is certainly crucial to the fate of the experiment, and if so, it is annoying to rely on such arbitrary setup.

* BC again, L. 231 states that the model is « driven entirely by buoyancy forces and there no imposed (non-zero) velocity or force boundary condition ». This is an biased description of the reality. Having free-slip is a very strong choice. It imposes zero horizontal velocity on each side, meaning that the stresses constantly have to adjust to the local stresses, at any time, to ensure this holds. This is a very common, yet arbitrary choice, that can't be ignored.

* This brings to a more general viewpoint, which is the fact that the model is therefore not self-consistent. The Earth is not a Cartesian box and other plates interact with mantle convection.

Miscellaneous comments:

- L.33-35: unclear. Perhaps you could reformulate, but I don't think this is the cornerstone of the paper anyways. I would perhaps recommend to cut down this aspect a bit.

- L. 80-82: I don't think this is quite right: the implicit force balance isn't quite right for it doesn't account for lithospheric buoyancy forces that actually balance those tractions in the one hand, and to the horizontal traction Σ_{xx} at the edge of the plate. See England and McKenzie 1982 for instance.

- L. 91-92: periodic folding: how relevant is that, in fact? Perhaps you could skip this part?

- L. 96-98: the fact that the timing is OK can't really be regarded as a validating proof. Timing scales with viscosity and can be tuned accordingly (within a certain limit).

- L. 111-114: it would be nice to see the streamlines on fig. 3.

- L. 123-127: in our 2012 paper (Husson et al., EPSL), we also show a latitudinal variation of the mantle drag, that increases towards the Central Andes from both ends. This might actually be viewed as supporting this results.

- L. 144-146: Trench migration is one of the favorite diagnostic tool to test plate tectonic models.

The current models seems to be at odds with observations, from fig. 4e, but this is not quite clear. How do they compare to, for instance, Ren et al.?

Regards,
Laurent Husson

Response to reviewers

Please find below my responses (in Arial blue) to the original comments from the reviewers (in Times black).

Reviewer 1:

Review Summary:

This paper presents a large-scale time-dependent three-dimensional model of subduction to test hypotheses for the formation of the Andes, magmatic arc migration, and variations in crustal thickness along the Andean margin in South America. The origin of the Andes and evolution of South America continues to be a hotly debated topic, despite the number of previous studies. As numerical modeling capabilities are advanced, more sophisticated hypothesis can be tested. This paper presents a significant contribution to the origin of the Andes. Most models of this scale do not examine thickness variations in the crust of the overriding plate over time, but rather examine dynamic topography as a proxy for mountain building. The approach here is thus a large step forward and raises provocative questions as to how to incorporate and interpret those effects into a large-scale 3D model. This paper will be widely read and received, and will have an impact across disciplines within geoscience, from the geodynamic modeling community, to field geology and tectonics, to igneous petrology, as well as the general community who studies the Andes. I recommend the paper for publication in Nature Communications. The detailed comments below include both conceptual comments on the model assumptions, additional relevant literature to address, as well as editing comments.

Main Text:

Line 27 – ‘whole mantle’ might be perceived as all of the Earth’s mantle, i.e., a global model. However, this is a regional model. What is meant here, is that the model depth spans the depth extent of the mantle. Please clarify.

Yes, correct, the depth extent of the upper and lower mantle. It has been clarified in the text.

Lines 38-40 – Mountain building has also been attributed to coupling between the upper plate with an underlying flat slab and/or lithospheric discontinuities, e.g., Jadamec et al., 2013 EPSL; Haynie and Jadamec Tectonics 2017.

A brief discussion on the role of flat slab/aseismic ridge/plateau subduction and overriding plate deformation has been added to the second paragraph of the main text, as well as a reference to Jadamec et al. [2013].

Lines 47-52 – Split sentence.

The sentence has been split in two.

Line 97 – The connection in time might be confusing. I understand that the difference is because the model time starts at 0. Perhaps better clarify for the non-modeler reader.

Has been modified to increase clarity.

Line 1113-1118 – Please comment on this large-scale mantle flow driving mechanism in the context of the recent results of Faccenna et al., 2017.

The paper by Faccenna et al. [2017], with its 2D numerical models of subduction, is along the same lines, and follows on from Husson et al. [2012], Faccenna et al. [2013] and the 2D numerical model of subduction in Schellart and Moresi [2013]. It is now incorporated, together with Husson et al. [2012], in several sections in the paper.

Lines 135-142 – Split sentence.

This part has been deleted (following comments from reviewer 2).

Lines 144-154 – This would be a good place to help the reader interpret the results in Figure 4. That is, that, the results as plotted indicate the curvature of the trench, with a landward apex in the centerline, is not due to a phase of trench advance. It is due instead to a reduced relative magnitude of trench retreat in the centerline of the slab, compared to the distal segments of the trench near the lateral slab edges.

Yes, indeed, good point. An explanation has been added to the section “Evolution of trench curvature”.

Also along these lines, either

here or elsewhere, it would be useful to clarify that the compression in the upper plate is not due to landward-directed flow of the upper plate or mantle (i.e., not toward the right, in Fig. 3), rather it is due to the gradient in trenchward motion within the upper plate (i.e., gradient in leftward upper plate motion in Fig. 3), due to the trenchward mantle flow.

Yes, also a good point. Some text has been added to the section “Mechanism of overriding plate deformation”.

Including an upper plate, and one with a relatively realistic lateral variations in lithospheric thickness is a significant contribution of this paper. Can the author comment on the overall significance of including an upper plate in these models resulting trench curvature and slab dip (e.g., Holt et al., 2016; Sharples et al., 2014).

We have already examined and discussed the role of an overriding plate in affecting the trench curvature and slab dip angle in earlier work [Meyer and Schellart, JGR 2013; Schellart and Moresi, JGR 2013], so I do not feel that the current work is the appropriate place to dwell on this again, also because it is somewhat of a technical aspect that will likely mostly be of interest to other geodynamic modellers of subduction (and of less interest to the wider community).

Please comment distinguishing the results here from other models examining slab geometry in South America (Martinod et al., 2005; Hu et al., 2016; and particularly, Faccenna et al., 2017).

Comparing the model results with the work of Martinod et al. [2005] is somewhat futile, as these authors presented generic subduction models focusing on aseismic ridge subduction and these models did not include an overriding plate, in contrast to the subduction model presented in the current work. So a comparison is rather outside the scope of the current work. We did include the work of Martinod et al [2013] in the second paragraph of the main text. The work of Hu et al. [2016] is also interesting, but it focuses on aseismic ridge subduction and flat slab subduction (not on overriding plate deformation), and is therefore not very relevant for the current work, with its focus on overriding plate deformation.

The paper by Faccenna et al. [2017], with its 2D numerical models of subduction, is along the same lines, and follows on from Husson et al. [2012], Faccenna et al. [2013] and the 2D numerical model of subduction in Schellart and Moresi [2013]. It is now incorporated, together with Husson et al. [2012], in several sections in the paper.

In addition, please comment on the expected effects of phase transitions (e.g., Bina et al., 2001; Arrendondo and Billen, 2016) on the slab dynamics, trench motion, and folding character of slab and trench motion, which in turn may also affect upper plate deformation in the model.

A minimum viscosity jump of a factor of 10^2 was applied between the upper and lower mantle. This simple implementation of the viscosity jump incorporates all the effects of the 660 km phase transition (viscosity changes, density changes and thermodynamic reactions from mineral phase transition) and captures the geodynamic essence of this barrier by reducing the sinking velocity of slabs in the lower mantle, as indeed implied by earlier numerical work on phase transitions and deep mantle mineral physics [e.g. Torii and Yoshioka, 2007; Ganguly

et al., 2009].

Additionally, more recent work has shown that studies in which a complete treatment of compositional layers and phase transitions is not implemented, the large-scale deformation of slabs is best approximated by a model with no phase transitions and no layers rather than including an incomplete approximation that over-predicts slab folding [Arredondo and Billen, 2016].

Part of the text from above has been added to the Methods section to address the reviewer's comment.

Lines 156-171 – It may be worth pointing out also that the steeper slab dip at the lateral slab edges is consistent with the overall observed pattern of steeper slab dips near lateral slab edges on earth (e.g., Lallemand et al., 2005).

Yes, indeed, good point. Has been added to the text.

Line 184-201 (and in methods)– This is somewhat of a new approach, and therefore additional text would help here or in the Methods. Specifically state either here or in the methods that Figure 8 shows the predicted change in thickness due to crustal shortening or extension, i.e., the vertical measurement of the bottom and top of crustal layers at the specified locations in the model.

I felt the most appropriate place to explain this is in the figure caption of Fig. 8, so I have added text there. Also note that Fig. 8 does not show the “predicted change in thickness, it shows the actual (measured) crustal thickness in the overriding plate at different times and places.

The discussion then, is about placing these numbers in context. Along these lines, there is no magma-genesis in the models. This should be specified in the methods.

This is now specified in the Methods section.

So, there it is understood that there is no contribution from the mantle in terms of accreting material to the base of the upper plate or within the plate through pluton solidification or a volcanic contribution. This will make it more clear that the results give bounds on tectonic contribution. Can the author comment on expected magma-genetic contribution from melting in the mantle & the addition of material to the upper plate from below?

No, I cannot really comment on this, as magma genesis has not been incorporated in the numerical model. This topic is also somewhat outside the scope of the current contribution.

Methods:

Line 208 – Add Underworld reference.

Has been added.

Line 209-210 – Is the Cartesian representation appropriate for a large swath of spherical earth. Please provide estimate why this assumption is appropriate.

We follow earlier works in which a Cartesian representation is used for numerical subduction models with large lateral domains, e.g. 7,000 × 7,000 km in Schellart et al. [2010], 12,000 × 12,000 km in Crameri and Tackley [2014], 8,000 × 3,000 km in Moresi et al. [2014] and 5,100 × 5,100 in Pusok and Kaus [2015]. This Cartesian representation is obviously a simplification of the spherical Earth, but earlier work implies that the subduction dynamics, subduction-induced mantle flow and the evolving slab and trench geometry are very comparable in both Cartesian and spherical models [Crameri and Tackley, 2014]. Furthermore, the *Underworld* code used for the simulation (version 1.6) did not include a capability for depth-dependent compressibility, and it was therefore decided to use Cartesian geometry.

Part of the text from above has been added to the Methods section.

Line 210 – Give number as well as ‘whole mantle depth’ (2900 km).

Has been added.

214 – The mesh doesn't solve the problem. I know the author knows this, the sentence is just awkward as written. The Eulerian part of the problem is solved on the mesh?

Text has been modified.

226 – Awkward at end of sentence. Try this "..., a maximum viscosity, $\eta_{UM(Max)}$, and a minimum viscosity, $\eta_{UM(Min)}$, such that $\eta_{UM(Min)} = 0.1\eta_{UM(Max)}$."

O.k. Text has been modified.

Thus, there is only a factor of 10 difference allowed between the maximum and minimum viscosity in the upper mantle. So, there is no more than 1 order of magnitude variation in the upper mantle viscosity. It would be useful to mention here or in the methods the scaled values, i.e., does this represent 10¹⁹ to 10²⁰ Pa s or 10²⁰ to 10²¹ Pa s? How might larger viscosity variations be expected to affect the result (e.g., Billen and Hirth, 2006; Jadamec, 2016).

To allow for reasonable (reasonably rapid) convergence in the numerical calculations the variation in upper mantle viscosity had to be limited to one order of magnitude. The current model took more than 2 years to run, and increasing the variation in upper mantle viscosity to two orders of magnitude or more would have resulted in additional years for the model to complete, and so make the whole modelling exercise unfeasible. In any case, I have added part of the text from above and a dimensionalized value for the maximum upper mantle viscosity ($\eta_{UM-MAX} = 5 \times 10^{20}$ Pa·s) to the methods section.

Larger viscosity variations might have resulted in somewhat faster plate velocities and slab sinking velocities, but its effect would likely be dampened by the higher shear tractions at the subduction interface, thereby not allow for major velocity increases to occur.

Line 232- What is the relative thickness of the three sublayers? Okay, I see this is in the model set-up figure (30 km, 30 km, and 20 km). Also specify that here?

Has been included.

Line 233-234 – Specify, assuming correlation between plate thickness and age from 6 space cooling model. Add reference, i.e., Turcotte and Schubert (2014) would suffice. How did the seafloor age vary for this region through time from plate reconstructions for the Nazca and Farallon plates? How would you expect this assumption affect the result, for example the younger ridge subduction (Sdrolias and Muller, 2008)?

Text on a half-space cooling model and references to Turcotte and Schubert [2002] and Cloos [1993] have been added.

Reconstructions of the oceanic lithosphere subducting along the South American subduction zone since 60 Ma show that the age of the oceanic lithosphere was mostly in the range 30-60 Ma [Sdrolias and Muller, 2006], which fits rather well with our own assumption. The age of the oceanic lithosphere at the trench in the period 200-60 Ma is much less constrained and rather speculative, and due to this lack of constraints it is kept at a constant thickness.

Part of the text from above has been added to the methods section to provide some justification for our choice of a 80 km thick subducting oceanic plate.

Also, there is no temperature evolution in the model. Please specify this in the methods to help the reader understand the model limitations and implications.

This is now specified in the Methods section.

How would you expect the slab morphology (width), thermal structure (density), and viscosity (temperature dependent weakening) to change over time due to diffusion and advection of heat in the mantle and warming of the slab?

The absence of thermal gradients in the model, and with it the absence of warming of the slab,

will not have a significant effect on the slab morphology, slab viscosity and slab-mantle density contrast in the upper mantle due to the relatively rapid rate of subduction (up to 7.5 cm/yr) and the slow rate of slab warming through conduction. Warming of the slab in the lower mantle would be more significant, and would likely produce a weaker slab segment in the lower mantle, which would thereby likely result in stronger slab folding producing tighter slab fold structures in the mantle. However, the density contrast of the lower mantle folded slab pile would not be significantly affected, as the warming of the folded slab segment coincides with the cooling of the entrained ambient mantle material in between the slab folds. As such, the thermal buoyancy contrast of the entire folded slab pile would not diminish and disappear on a timescale of 100-200 Myr, and so the driving mechanism of the lower mantle slab would not be significantly affected.

The text from above has been incorporated in the methods section.

Similarly, the model simulates subduction for a sustained amount of time ~200 My. How might the thermal evolution be expected to affect the continental growth and possibly continental delamination or drips of mantle lithosphere from the overriding plate over time?

Delamination/dripping are second/third order effects of the overriding plate that will not affect the main outcomes of this work as presented in Figs. 3-9. Furthermore, it is not possible to simulate such smaller-scale processes in a subduction zone model of this size with the present resolution. It would require a much higher resolution (order of magnitude?) that is currently not feasible (considering that the current model took more than 2 years to complete).

Lines 235-236 and subsequent parts on viscosity – It would be useful somewhere to dimensionalize the viscosity. If η_{UM} is 1020 Pa s, then the viscosity of the upper and middle layer of the subducting plate would be 1023 Pa s, and the viscosity of the forearc and backarc would be on the order of 1022 Pa s. Alternatively if the η_{UM} was 1019 Pa s, the oceanic and continental plate viscosities would be on the order of 1022 Pa s and 1021 Pa s, respectively. It would be useful to compare estimates for upper plate viscosity based on observations (e.g., England and Molnar, 1997) and theory (Jadamec et al., 2007), albeit these references are for the Tibetan plateau.

The viscosity of the upper mantle is now dimensionalized (second paragraph of the Methods section) and ranges from 5×10^{19} to 5×10^{20} Pa·s. These values fall within the estimated sub-lithospheric upper mantle viscosity range (10^{19} - 10^{21} Pa·s) in nature [Artyushkov, 1983; Ranalli, 1995], while $\eta_{UM(Max)}$ is close to the estimated average sub-lithospheric upper mantle viscosity ($3\text{-}4 \times 10^{20}$ Pa·s) as deduced from glacial isostatic studies and mantle convection studies [Lambeck et al., 1998; Mitrovica and Forte, 2004]. It was decided to not compare our upper plate viscosity values with those of the Tibetan Plateau, because the Tibetan Plateau is located in a very different geodynamic setting (a continent-continent collision zone) compared to the South American subduction zone setting.

Line 251- “upper layer representative of the crust” rather than ‘crustal layer’. Or is there elasticity to this layer? What equation governs its rheology?

All the domains of the overriding plate have linear viscous rheologies. This is now emphasized in the text.

Line 276 - Isn't this 72 km, 105 km, and 173 km, respectively (Syracuse and Abers, 2006). Please clarify.

The values 72 km, 105 km and 173 km are uncorrected values for the minimum, mean and maximum depth to the top of the slab (i.e. not taking into account hypocentre errors). When such hypocentre errors are taken into account, then the values are 83 km, 125 km and 208 km [Syracuse and Abers, 2006], and these are the values used here. This is now specifically stated in the Methods section.

Lines 262-263 – Specify the total number CPU hours for the model as well (e.g., 496 CPUs * X hours).

It ran for ~3000 hours on 496 CPUs, so this is about 1.5 million CPU hours. This has been added to the methods section.

Lines 278-281 – Why are only references for the Jurassic arc given here? What about constraints for the younger arc?

The location for the present-day magmatic arc as plotted in Fig. 1a is based on Syracuse and Abers [2006]. This is now explained in the Methods section.

Line 283 – Please clarify based on comments for Lines 184-201.

See response to comments on Lines 184-201.

Figures:

Figure 1 – Is the horizontal axis for c same as in a? If so, add ticks. If not, why not? Is the trench normal distance for Peru through time not constrained?

Unfortunately the trench-normal distance for arc magmatism in Peru has not been constrained in Jaillard and Soler [1996]. So tick marks cannot be added.

Figure 2b – If I understood the methods correctly, then in the upper mantle (yellow layer in the larger domain image) specify: $\eta_{UM}(\text{Min}) < \eta_{UM} < \eta_{UM}(\text{Max})$

Yes, indeed. Has been added to Fig. 2b.

Figure 3 – Specify v' is non-dimensional velocity.

Explanation has been added to the figure caption.

Figure 4a – This is a conceptual/understanding of results comment. I added it also in the comments where Figure 4 is referred to in the text. The blue line in Figure 4a is positive the whole time (always retreating). In Figure 4e there is a trench curvature change, such that the final curvature at time 182.6 has a landward apex in the center. This occurs even though the trench along the centerline retreats the entire time, in absolute velocity. Thus the landward directed trench apex is a result of relatively less retreat along the centerline than at the edges (not due to trench advance in the centerline region). It might be useful to point this out, because the curvature might be thought to be due to trench advance along the centerline.

Yes, indeed, good point. An explanation has been added to the section “Evolution of trench curvature”.

Figure 4a – What is the significance of a negative subducting plate velocity (black line) in the very early phase of subduction (first 18 million years of model)? Is that a numerical artifact or is there something pulling the surface part of the subducting plate backwards (away from the trench) in the early phase?

The negative subducting plate velocity in the first ~16 million years of the model run are not significant considering that they are only of the order -2 mm/yr to -1 mm/yr. It might be a result of the steepening of the initial slab perturbation, which is initially dipping at ~30°.

Figure 4a,b,c – The connection and between the upper plate deformation measurements in (a) and (b) are not clear. The signal changes from positive to negative at about 70 Ma in (a), but not until about 140 Ma in (b). I know these represent different quantities, but is not clear intuitively how the values in b,c relate to the deformation rates in 4a.

The green curves in Fig. 4a show the (instantaneous) deformation rates in the forearc and backarc in the centre of the subduction zone, while the black curve in Fig. 4b gives the finite deformation. So the slope of the curves in Fig. 4b gives the (instantaneous) deformation rate, with a positive slope indicating an extension rate, and a negative slope indicating a shortening

rate.

Text has been added to Figure caption 4 to explain that the slope in Fig. 4b gives the (instantaneous) shortening rate.

Figure 5 – This is a nice way to show the relative position of the trench and slab in a mantle depth slice.

Yes, thanks.

Figure 6 – This is somewhat of a cosmetic comment, but might make this set of compilation figures easier to understand at first glance. Use two of the σ_{xx} scale bars – one under the upper right in left column and the other in upper right of right column. It will make it easier to see at a glance that the entire left column is results on one timestep and the right column is the same set of results but at a later timestep.

Yes, good point. Has been modified.

Figure 6b – Specify which direction is positive (toward or away from trench).

Towards the right is positive. This is now specified in the figure caption.

Figure 7c – It is very hard to tell the two dark red lines apart (the centerline and 2400 km away from centerline lines).

The colour contrast has been increased in the revised figure.

Figure 8 – This is a very significant contribution. Very few large-scale 3D numerical models of subduction quantitatively predict crustal thickness variations in upper plate over time. See questions in earlier part of review to clarify how this plot relates to the discussion.

See earlier responses.

Figure 9c – It is really hard to distinguish the two domains.

Transparency of the orange domain (representing the arc width in the centre of the subduction zone) has been increased such that the hashed domain (representing the arc width at 2400 km from the centre of the subduction zone) is now better visible.

Reviewer 2:

Reviewer #2 (Remarks to the Author):

Schellart presents an intriguing model linking the early extension (~ Jurassic to early Cretaceous in age) and the subsequent switch to compression to early upper mantle subduction (backarc extension), followed by whole mantle subduction to produce the Andean orogeny. As stated in the abstract the strength in the model is that it does reproduce the maximum shortening and crustal thickness observed in the Central Andes and their progressive northward and southward decrease. The downside of the model that is not acknowledged anywhere in the manuscript is that it hugely (hugely) over predicts the magnitude of extension both the limited extension in the central Andes and the more prevalent extension in the southern and northern Andes. And thus also over predicts the amount of shortening in the southern Andes (by an order of magnitude) While I do not think that this is a reason to not publish the paper or the model (in this journal), I think it is disingenuous and limiting to not acknowledge that problems exist and not discuss why (why the over prediction, mis-prediction). Thus the short version of this review is that I think the manuscript should be published in Nature Communications but I also think that the author needs to acknowledge the problems and thus discuss potential solutions.

Sure, existing problems should be acknowledged and potential explanations should be discussed. Please see my responses below.

Most of what I have added below addresses matching the model to data.

Magnitude ---To give an idea about the magnitude, the Basin and Range in the western US has undergone ~250 km of extension. This extension has produced metamorphic core complexes that rapidly exhumed basement, produced multiple (10-20) normal fault bounded basins that have accumulated 3-6 km thick Tertiary basin fill during this extensional period. (~ 35-40 Million years). 250 km is the low amounts of extension that is modeled in the Central Andes where there are limited structures and sedimentary rocks that would support extension. (This is displayed in Figure 1 on the paper). Moving from south to north there are extensional basins as far north as 20° S, and a support for extension in rocks preserved along the coast starting again at 15° S. There is really nothing, particularly in the folded and faulted rocks that comprise the central Andes between 20 and 15, that support anything but really limited (a few kms) of Jurassic – early Cretaceous extension in that region.

While there is a lot to support extension of that age farther south (such as Salta Basin, central Chilean basin, Neuquen Basin) that any of these basins accommodated the minimum 250 km and the proposed 400 extension is a bit far fetched. The amount of continental extension that preceded the opening of the Atlantic Ocean ranges from 60 to 600 km, with many segments falling into the 200-400 km range of extension suggested here (point is with that much extension an ocean basin opens). Extension creates basins and preserves sedimentary rocks and those rocks get caught up in the subsequent compressional fold thrust belt. Thus it is not possible to just dismiss the lack of measured extension values with arm waving that the basins were inverted, eroded, “hidden” by later compression etc. The rocks that accumulated from that period would have been incorporated into the system and there has not been that much erosion in the Andes. There large thermochronometer data sets that allow us to track the thermal history of the rocks that are there and the surface rocks have not been buried by large sedimentary basins that have been erosionally removed. The point of this paragraph is to argue (strongly) that the extensional magnitudes presented in the model are wrong. Why? And what does that imply about our understanding of the system.

There is much published about shortening in the Andes and there exists a significant number of quantitative estimates of such shortening, and its variability along the Andes. One problem (for me) has been that, although there are a number of papers that discuss the period of Jurassic-Early Cretaceous extension along more or less the entire length of the western margin of South America preceding Andean shortening, there are (as far as I am aware) no quantitative estimates of this extension. There is one review work that I am aware of [Maloney et al., 2013] (and this work is discussed in the paper), which convincingly argues for larger amounts of extension in the north and south (with the formation of backarc basins), and reduced extension in the centre. Although such trends are consistent with the model results, no quantitative estimates of extension are provided. From the reviewer’s discussion above it appears very plausible that the early phase of extension in the Central Andes domain was rather limited, maybe up to a few tens of km, so about an order of magnitude less than what was produced in the model (~250 km). The amount of extension in the northern Andes (Colombian Marginal Seaway) and Southern Andes (Rocas Verdes Basin) is likely much larger, of the order of a few hundred km, and so of the same order of magnitude as the extension near the lateral slab edges documented in the models. The large extension in the north and south is implied by the (backarc) ocean floor rocks that have been documented in these regions and point to the former existence of major backarc basins in the north (Colombian Marginal Seaway [Maloney et al., Tectonics 2013; Villagomez et al., Lithos 2011]) and the south (Rocas Verdes Basin [Maloney et al., Tectonics 2013; Dalziel et al., Nature 1974]).

A potential explanation as to why the numerical model significantly over-predicts the amount of extension in the Central Andes is the adopted initial overriding plate set-up, with a relatively thin forearc+backarc region (60 km thick) with a relatively weak backarc rheology (with a viscosity that is $100\eta_{UM-Max}$). This set-up is very reasonable for an overriding plate that is experiencing active subduction [Currie and Hyndmann, JGR 2006], but it might be too thin

and too weak for the first several tens of millions of years after subduction initiation, so for most of the Early and Middle Jurassic. A stronger and thicker backarc would result in much reduced backarc extension rates. For example, a viscosity increase by an order of magnitude would decrease the extension rate (and the extension) by an order of magnitude. If such high viscosities prevailed in the first ~40 million years of subduction, then backarc extension would be reduced by an order of magnitude, which appears more plausible, but backarc extension and spreading in the north and south would still be of the order of several hundreds of km, because the extension continued for much longer here, and could thus take place in a thermally thinned and weakened backarc region.

In any case, part of the text from above has been incorporated in the section “Subduction evolution, mantle flow and overriding plate deformation”.

The other problem with having too much extension is that then there is too much compression need absorb the excess crust and come up with modern thickness and shortening values. However particularly north and south of the central Andes these shortening magnitudes are orders of magnitudes higher than documented. Again for example the Neuquen Basin in Argentina (Horton et al., 2016 cited in the paper) gives shortening estimates of 15 -30 km (documented in the inverted basins on the eastern side). The model argues for 300 km of compression.

Following a suggestion from reviewer 2 below to plot the observed deformation with the modelled deformation together (see revised Figs. 4c and 4d), one can see that for the Schepers et al. [2017] curve, when an additional ~200 km of western cordillera shortening is added, there is a rather nice fit (both in terms of shape and magnitude) between this curve (black dashed curve in Fig. 4d) and the model curve for 182.6 Myr (blue curve in Fig. 4d).

The increased shortening close to the lateral slab edges could possibly be explained by the underthrusting/subduction of the backarc oceanic basins of the Rocas Verdes Basin and Colombia Marginal Seaway.

In any case, part of the text from above has been incorporated in section “Subduction evolution, mantle flow and overriding plate deformation”.

The 500 km of compression proposed for the central Andes is in the ballpark (particularly when accounting for the hard to document but certainly present shortening in the western cordillera that *is* Cretaceous in age and mostly covered by modern volcanics)

Yes, agreed.

I think it would be insightful to all thinking about these processes to demonstrate not only the parts that agree but to highlight the misfits and the misfit magnitude.

Yes, agreed, see responses above and below.

Rotation --- Another interesting implication. How does it match with measured values? This can be done (easily enough), emphasize the magnitude of whole lithosphere rotation produced in the model, its age, and compare it to measured rotations of the CARP. Versus simply saying CARP argues for counter clockwise rotations north of the bend and clockwise south and look the model produces that too. Without doubt, the measured rotations are subject to local structural influences, have large uncertainties, and were sampled from lithologies older than Late Cretaceous (thus may record rotations unrelated to Andean deformation). But, you can also focus on paleomagnetic data from synorogenic sediments located in the Subandean region (compiled by Roperch et al., 2006; republished in Eichelberger and McQuarrie 2015) and highlight the magnitude that would be occurring from 15 Ma to present (south limb: 12°; north limb: -6°).

Yes, interesting. The work of Cobbold et al. [2007] presents Jurassic-Cretaceous rocks with anticlockwise rotations of ~5-70° north of the Arica bend and clockwise rotations of ~0-50° south of the bend, which is consistent with the anticlockwise and clockwise rotations north and south, respectively, of the orocline bend in the model. However, maximum rotations of 16° observed in the model (see rotation values that I've added to Fig. 4e) are on the low side compared to nature. This might be partly due to the gentler curvature of the orocline in the

model, possibly resulting from the lack of brittle rheologies in the model, compared to the more acute curvature in nature, and partly due to local block rotations in nature (and the absence of such block rotations in the model).

I've added part of the text from above to the section "Evolution of trench curvature and trench migration".

Crustal Thickness --- Any argument of amount of shortening and resulting crustal thickness is based on the assumption of original crustal thickness. The Kley and Monaldi curve shown in figure 1d assumes an initial (pre-shortening) thickness of 40 km. Decreasing the initial crustal thickness will of course require more shortening to account for the modern crustal thickness and have implications for amount of material available for parts of the Andes with less shortening and implications for recycling of crust and lower crustal flow. Saying the original crustal thickness of the Andes is 40 km is quite generous and was used by Kley and Monaldi back when shortening estimates were significantly lower than what was needed to explain modern thicknesses. We used it as a way of including the thickness of tertiary foreland basin rocks (5-7 km) on top of an original 35-33 km thick crust (Eichelberger et al., 2015, EPSL). But also show a 35 km version and a variable thickness version with eastern Cordillera thicknesses as low as 30 km (10-15 km sedimentary rock thickness and 15-20 km of basement). Again because of the preserved 12 km thick Paleozoic sedimentary rock record across the fold and thrust belt where the shortening estimates are constructed—it is hard to argue for 250 km of extension. This is $>1/4$ of the original length of the restored cross sections (~800 km) and hard to do without leaving a trace.

Yes, I agree that the final crustal thickness depends on the assumption of the original crustal thickness. A 30 km thick initial crust was chosen, which is reasonable for the Southern and Central Andes, considering the present-day far-field crustal thickness (30-35 km), but on the low side for the Northern Andes (35-40 km) [Chulick et al., 2013]. In any case, I have deleted the last 8 lines in the section "Crustal thickness evolution", as it is likely that this explanation does not apply anymore, considering the model overestimation of early extension in the Central Andes.

What is important to note, from a geodynamic point of view, is that the driving mechanism of Cordilleran mountain building is capable to both deform the overriding plate lithosphere and support the major buoyancy forces that come with 74 km thick continental crust.

Part of the text from above has been incorporated in the section "Crustal thickness evolution".

As a clarification any balanced cross section takes into account shortening east of the volcanic arc. For Bolivia the highest estimates are 350 km of shortening, with that increasing to about 400 km of shortening taking into account map view shortening (Eichelberger and McQuarrie 2015, GSAB). The shortening estimates in McQuarrie et al, 2005 that are 530-580 km of shortening state that balanced cross sections account for 330 km but also postulate ~200 km of shortening in the western cordillera that predates shortening to the east to account for preserved foreland basins. We do know that there is some magnitude of Cretaceous crustal shortening there, However we do not know the extent. The amount necessary to produce the basins preserved in Bolivia is 150-200 km. If the system switches to compression at 80 Ma, there is 30 Ma to deform the western Cordillera before deformation is solidly in the eastern cordillera (Rak et al, in revision) so the rates are ~ 5-7 mm/yr (which work). This additional amount can also get added to the systems in the south but that only increases the permissible shortening there from 50 km to 250 km—still short from in the 300 in the model but getting closer.

Of course the larger mismatch problem is farther south (and north) where the shortening estimates are much, much less than that needed to produce the measured crustal thickness. (as highlighted in Kley and Monaldi) the shortening estimates are more refined now but problem north and south of the Bolivian Andes remain. (e.g. Eichelberger et al., 2015) it is always possible for some model to predict the amount of shortening necessary to produce the right thickness. The problem is –did that amount of shortening actually occur.

As noted earlier, following a suggestion from reviewer 2 below to plot the observed deformation with the modelled deformation together (see revised Figs. 4c and 4d), one can see that for the Schepers et al. [2017] curve, when an additional ~200 km of western cordillera

shortening is added, there is a rather nice fit (both in terms of shape and magnitude) between this curve (black dashed curve in Fig. 4d) and the model curve for 182.6 Myr (blue curve in Fig. 4d).

The increased shortening close to the lateral slab edges in the model could possibly be explained by the underthrusting/subduction of the backarc oceanic basins of the Rocas Verdes Basin and Colombia Marginal Seaway.

In any case, part of the text from above has been incorporated in section “Subduction evolution, mantle flow and overriding plate deformation”.

Evolution of dip angle and subduction erosion--- So for me, the argument for subduction erosion has always been the occurrence of the Jurassic arc rocks *at* the modern coast and within 25-75 km of the modern trench. So instead of simply arguing slab angle can solve the problem completely (it really can not viably explain the arc rocks (plutons) 25 km from the modern trench) what is the minimum amount of subduction erosion needed to account for that (taking into account model dip angles)

The smallest separation between the coastline and the trench in Southern Peru and Northern Chile, where Jurassic magmatic arc rocks are documented in the coastal regions, is ~74 km (\pm a few km). The smallest separation between the trench and predicted magmatic arc in the centre of the subduction zone in the geodynamic model is 58 km (Fig. 9c, at a time of ~20 Myr). So in principle, the model can account for the location of the Jurassic arc observed in nature that would have formed in the early stage of subduction. It is certainly possible, however, that there has been some subduction erosion along the west coast of South America of several, or possibly several tens of kilometres. Nevertheless, the geodynamic model does not require any subduction erosion, as it can explain all the present-day separation between the trench and the Jurassic arc, as well as the trench and the active arc.

Part of the text from above has been incorporated in the main text in the section “Continental crustal destruction or growth”.

Parts of the model that I really like: I think it is intriguing that even with the over prediction on the south (and possibly over prediction in the north) the difference in predicted shortening predicted in the central part of the model and that predicted to the south is ~ 300 km – similar to the measured differences (350-400 in the central Andean plateau and ~ 500-100 to the south). I am intrigued by the resulting predicted variation in shortening rates (and the cyclicity of them) cyclicity in Andean shortening was proposed by DeCelles et al., 2009 (Nature Geo) albeit the driver they propose is different. We also found cyclicity in shortening rates in the central Andes (Rak et al, in revision, tectonics). The timing of when extension and shortening happens is really good (even though magnitude could be way off).

Yes, the difference in shortening between the centre and north, and between the centre and south, as reproduced in the models, and observed in nature, is intriguing. A comparison between model and nature is now facilitated through the modification of Figs. 4c and 4d, where shortening curves are plotted as derived from the numerical model and derived from observations in the Andes.

The cyclicity in overriding plate deformation is also intriguing. I have added a reference to DeCelles et al. [2009] and Jailard and Soler [1997] who have proposed such cyclicity for the Central Andes in the section “Subduction evolution, mantle flow and overriding plate deformation”.

Clearly this is a long review due to an obvious interest in the paper and its implications. It definitely caught my interest and I enjoyed reading the paper and thinking about these problems. My opinion is the paper will be stronger, and more useful if the author acknowledges both the fits and the mis-fits and presents ideas for what are driving the misfits in the model (not just attribute it to uncertainty in the data—there definitely is uncertainty but that uncertainty will not get you 250-400 km of extension!)

Small points:

Line 27 I think “demonstrating” is too strong.
Has been reworded.

Figure 1d I think there is a way to make this more interesting and more useful. I would also cite Schepers et al., 2017 for this because although their curve is not that far off of Arriagada, it is based on actually shortening data (Eichelberger et al., and updated shortening references north and south) that are not in Arriagada (theirs is model based). I thought that Schepers actually published the data for this in a supplementary info file (but can not find it). You can also put a dashed line above the Schepers line accounting for the ~ 200 km of shortening that is documented but super hard to quantify in the western cordillera (verses just a point). (citing McQuarrie et al., 2005 and Rak et al., 2017) and postulate this exists all along the coast (maybe??). What I think would be interesting and helpful is to plot your model predicted shortening over the actual shortening estimate lines (either in this figure or another figure)

Yes, nice suggestions. I've added the curve of Schepers et al. [2017] to Fig. 1d. I've also plotted curves for the shortening estimates based on crustal thickness and balanced cross-sections in Figs. 4c and 4d, respectively. As can be seen, the model curve for 182.6 Myr and the crustal thickness curve from Kley and Monaldi [1998] in Fig. 4c are reasonably comparable. A good fit (in particular in terms of shape, but also magnitude) is obtained between the model curve for 182.6 Myr and the Schepers et al. [2017] curve with the added ~200 km of western cordillera shortening postulated for the entire coast (see blue line and dashed black line in Fig. 4d).

Sorry, but I cannot find Rak et al. [2017]. I guess it is still in review.

As stated previously it is important to state that the line based on crustal thickness is assuming a 40 km thick initial crust. (This of course can get replotted assuming any initial crustal thickness).

The figure caption of Fig. 1 now states the assumption of a 40 km thick crust for the curve of Kley and Monaldi [1998].

Figure 2. It took me zooming in on this figure to find out the initial crustal thickness was 30 km (as I state above I think that is the thinnest one could go for initial Andes). The modern crustal thickness of the foreland basin is 40 km and has 3-6 km of foreland basin sediments on it and about 7 km of Paleozoic rocks. Thus 30 km basement is bare minimum. One could assume the basement thins and sediments increase to the west and that it stays 30 km. Particularly if you are discussing crustal thickness evolution you should state the initial crustal thickness (I know it is on figure 2 and 8 now but I felt I had to dig for it) and how it varies spatially (figure 8 a is great for the center but I think that plot at 1200 km out and 2400 km out would be insightful as to potential problems).

Fig. 8 now states that the original continental crustal thickness at the start of the model is 30 km.

The present-day thickness of the South American continental crust in the (undeformed) far backarc is ~30 km for profiles D and E in the south, ~35 km for profile C in the centre and ~40 km for profiles A and B in the north. So, yes, I took a lower-bound value for the initial crustal thickness. This is now mentioned in the text in the section “Crustal thickness evolution”.

Not sure what is meant with the last point, as I do include crustal thickness plots for 1200 km and 2400 km from the centre, as shown in Fig. 8b.

best regards

Nadine McQuarrie

Reviewer 3:

Reviewer #3 (Remarks to the Author):

Review of Nature Communications manuscript NCOMMS-17-13241, by Wouter Schellart, entitled « Andean mountain building and magmatic arc migration driven by subduction-induced whole mantle flow ».

Laurent Husson, July 4th, 2017

Wouter Schellart presents a numerical model that aims at explaining the principal features of the Nazca - South America convergence: slab dynamics, mountain building, trench geometry, magmatism etc... all diagnostic processes that have been examined previously in numerous studies, but perhaps not in such a comprehensive manner. The main outcome of the model is to reproduce most of these feature using a very simple model (yet, simple doesn't necessarily mean too simple), that nevertheless is itself an impressive -and expensive- numerical performance (2 years computation time on 496 processors!). This work is remarkable in that sense.

From a scientific perspective, this model follows a very long series of cartesian subduction models, that present the advantage of delivering clear and easily understandable messages, but the drawback of not being fully consistent; the Earth is not cartesian, it is the thermal convection of the Earth that drives plates, and plates constantly interact together. Both methods are insightful, but it may be critical in many cases, as it is for the Andes (as proposed as early as 1998 by Lithgow-Bertelloni et al., Science).

Something that limits the attraction power of this paper is the fact that these results build on very many earlier studies, including some by the author himself, and is in that sense not so innovative. One merit is perhaps to provide a unifying view of such prior studies, but in many cases the conceptualization is not new. Examples are:

- An important result is the transition between upper mantle driven convergence and whole mantle driven convergence: the shallow slab excites a small size convection cell that promotes slab rollback, while the deep slab conversely drives major convection cells that promote the development of large orogens.

This is the core of a paper by Faccenna et al, 2013 (Tectonics, not cited), although they do not prove it experimentally.

The key point here is that they do not prove it experimentally, as the reviewer notes, so they do not demonstrate the physical viability of their proposals. A follow-up on this paper is the very recent one by Faccenna et al. [2017], in which they present 2D Cartesian subduction models to demonstrate their proposed mechanism. This new paper is now referred to and discussed in several places in the main text.

- Another important is the fact that the upper plate could be dragged westward by the return flow of the subjecting slab. This is already explored and quantified by a force balance, by Wdowinski and O'Connell, 1991 (not cited).

The paper by Wdowinski and O'Connell [1991] presents kinematically driven 2D numerical models of upper mantle subduction, finding that the return flow drives overriding plate shortening. This is entirely opposite to more recent findings [Schellart and Moresi 2013] of (more realistic) dynamic numerical models of upper mantle subduction in 3D space, showing that the return flow drives overriding plate extension (not shortening), and the findings presented in this work, showing that whole-mantle-depth return flow is required to drive crustal shortening and major orogeny in the Andes. Nevertheless, I have added this reference to the second paragraph of the main text in which a number of previous works are briefly discussed that have proposed that subduction-induced mantle flow drives overriding plate deformation.

- Trench curvature associated to the westward migration of the the slab is proposed by Russo and Silver to (1998?), followed later by Schellart et al (2007).

I'm guessing the reviewer refers to Russo and Silver [1994], who proposed a conceptual model for the development of the trench curvature along the South American subduction zone (based on seismic shear wave splitting in the mantle), relating it to mantle flow. This interesting work did not ascribe the curvature development to the large size of the subduction zone, though, which is what Schellart et al. [2007] proposed and demonstrated with numerical subduction models. The most significant simplification in the models of Schellart et al. [2007], however, was the absence of an overriding plate. One of the novelties of the current work is the inclusion of an overriding plate in the numerical model, such that trench curvature, overriding plate curvature, overriding plate deformation and crustal thickening could be studied in concert. An additional novelty is that the current model spans the entire age of the subduction zone, some 200 million years, thereby showing the evolution of the trench and overriding plate for this major time span.

- Many of the current results also belong to Capitanio, or Uyeda and Kanamori (1982), or Martinod et al., to cite a few.

The work of Capitanio et al. [Nature 2011] on the Andes is briefly discussed in the manuscript, but these authors look at vertical stresses to infer topography in the overriding plate, and ascribe the variation in topography to the variation in subducting plate age. These authors do not investigate overriding plate shortening or extension, nor do these authors investigate the spatial link between mantle flow and overriding plate deformation, nor the long-term (200 million year) evolution of the subduction zone, nor the evolution of the slab dip angle and the migration of the magmatic arc, nor the evolution of crustal thickening.

As far as I am aware, there is no paper Uyeda and Kanamori [1982].

The papers from Martinod et al. are certainly interesting, but the ones that focus on subduction and/or the Andes are generally concerned with the subduction of aseismic ridges and plateaus [e.g. Martinod et al., GJI 2005, EPSL 2010, Tectonophysics 2013] and their local effect on the slab dip and/or overriding plate deformation. The current work focusses on the entire South American subduction zone not on local effects of subducting aseismic ridges. Nevertheless, the paper from Martinod et al. [2013] is now briefly discussed in the second paragraph of the main text.

- Presentation of earlier models (L. 56-58) is partial and doesn't give justice to the successfulness of earlier attempts. Of course, this is not the place for a review, but many other models exist, some explaining most features just as well. I can for instance refer, with some embarrassment, to Husson et al., 2012 (EPSL), which is in the same lines as the current paper, with a noticeable difference that upwellings are thought to be primordial, unlike in the current study, which is interesting.

Indeed, this is not a review paper, so discussion of all earlier attempts is not possible. The original version discussed 5 different models in the paragraph referred to above [Molnar and Atwater, EPSL 1978; Capitanio et al., Nature 2011; Silver et al., Science 1998; Sobolev and Babayko, Geology 2005; Lamb and Davis, Nature 2003]. In the revised version, several additional papers are now included and briefly discussed [Sleep and Toksoz, 1971; Wdowinski and O'Connell, 1991; Schellart et al., Science 2010; Husson et al., EPSL 2012 and Faccenna et al., 2017]. A reference to the proposed primordial upwelling [Husson et al., 2012] is now also included in the section "Mechanism of overriding plate deformation".

- Model limitation:

* perhaps one strong limitation are the boundary conditions around the South American plate. At the beginning of the experiment, the plate touches the right end of the box, where BC are free slip (and hence no horizontal velocity is permitted). It takes some work to make it move to the center of the model. This is expressed as a resistance to displacement, and thus, adds up an extensional force in the upper plate. This force gradually disappears as the plate moves to the center of the box, and this occurs when compression occurs too. Could this be quantified?

The reviewer is incorrect here. The overriding plate does not touch the right-hand sidewall of the box. As can be seen in Fig. 2, there is a 50 km separation between the trailing tip of the

overriding plate and the sidewall. What's more, the trailing edge of the overriding plate is tapered, such that the point where the overriding plate reaches a constant thickness is located 200 km from the lateral sidewall. As such, there is no significant force at the sidewall in the vicinity of the overriding plate trailing edge.

* free-slip at all boundaries (L. 229). I assume this is also true at the surface. I'm a little out of date on these aspects, but I feel that many recent papers have shown the force balance to be different with free surface and free slip, hence different crustal growth scenario. How important is that?

I'm not aware of any study in which it has been tested how the crustal growth in the overriding plate of a subduction zone might be affected by either using a free-slip top surface or a free surface (using the "sticky air" method). It has been shown that a free surface facilitates the formation of single-sided, asymmetrical, subduction in global mantle convection models, but this is not an issue in the current work, as the simulation clearly shows single-sided asymmetrical subduction throughout the 200 Myr evolution of the model.

* The setup / initial conditions are very important. Choice is made to have a 800 km wide low viscosity back-arc, 60 km thick only, with no justification. This is certainly crucial to the fate of the experiment, and if so, it is annoying to rely on such arbitrary setup.

The geometry and set-up of the overriding plate is based on the detailed work of Currie and Hyndmann [JGR 2006] on the thermal structure of subduction zone forearc-backarc regions of 10 ocean-continent subduction settings in the Pacific domain using observational constraints. They found comparable geometrical set-ups for these settings, generally consisting of a cold, ~200-300 km long, forearc, a warm, ~600-1000 km long, backarc with a lithospheric thickness of ~60 km, followed in a number of cases by a cold, strong and thick cratonic lithosphere in the far backarc implying a lithospheric thickness of 150 km or more. Part of the text from above has been added to the Methods section to provide a justification for the set-up of the overriding plate.

* BC again, L. 231 states that the model is « driven entirely by buoyancy forces and there no imposed (non-zero) velocity or force boundary condition ». This is an biased description of the reality. Having free-slip is a very strong choice. It imposes zero horizontal velocity on each side, meaning that the stresses constantly have to adjust to the local stresses, at any time, to ensure this holds. This is a very common, yet arbitrary choice, that can't be ignored.

I do not see how the use of free-slip lateral boundary conditions and the absence of externally imposed (non-zero) velocities are a "biased description of the reality". In numerical regional subduction models, free-slip lateral boundaries are the most frequently used boundary conditions, and they are used in an attempt to minimize the effect of these lateral boundaries on the model outcomes. Other well-known boundary conditions are no-slip and periodic, but these generally have a larger influence on the model outcomes.

Additionally, the effect of the lateral sidewalls on the model outcomes in the current work is very small due to the very large lateral extent of the model box (10,000 km by 6,000 km) and the large separation (many thousands of km) between the subducted slab and the lateral side-walls. In earlier work we have tested the influence of lateral side-walls on subduction dynamics using a free-slip boundary condition [Schellart et al., Nature 2007; Schellart et al., Science 2010]. These earlier works showed that the separation of the slab and the lateral side-walls should be ≥ 0.5 times the width of the slab for these side-walls to not significantly affect the evolution of subduction and mantle flow. Another condition is that the separation distance should be larger than the thickness of the mantle reservoir (2900 km in the current model). These two conditions are clearly met in the numerical model presented in the manuscript (see Figs. 2 and 3).

Thus, following Stegman et al. [G-cubed 2006], Schellart et al. [2007, 2010], Moresi et al. [Nature 2014] and many others, we choose free-slip boundary conditions along the side-walls, which minimizes the influence of these boundaries on the flow in the system.

Part of the text from above has been included in the methods section to support the choice of free-slip lateral boundary conditions.

* This brings to a more general viewpoint, which is the fact that the model is therefore not self-consistent. The Earth is not a Cartesian box and other plates interact with mantle convection.

Not sure what makes reviewer 3 to conclude that the model is not self-consistent. If it is the usage of Cartesian geometry, then the reviewer is incorrect, as both Cartesian and spherical geometries allow for self-consistency. Indeed, the fact that we use a buoyancy-driven, closed-system approach indicates that the numerical model is self-consistent. Sure, the Earth is not a Cartesian box, but this is a model simplification (not an indication of lack of self-consistency), used previously by myself and colleagues, and many others, e.g. Husson et al. [Tectonophysics 2012]. Text has been added to the Methods section following a comment from reviewer 1 in relation to the simplification of using a Cartesian geometry. It is also true that other plates interact with mantle convection, but the current geodynamic model is a regional subduction model. It is common practice for regional subduction models to only include one plate (the subducting plate) or two plates (the subducting plate and overriding plate), and to study their effect on subduction, mantle flow, plate and slab deformation in isolation.

Miscellaneous comments:

- L.33-35: unclear. Perhaps you could reformulate, but I don't think this is the cornerstone of the paper anyways. I would perhaps recommend to cut down this aspect a bit.

I have modified the last two sentences of the abstract slightly to improve clarity.

- L. 80-82: I don't think this is quite right: the implicit force balance isn't quite right for it doesn't account for lithospheric buoyancy forces that actually balance those tractions in the one hand, and to the horizontal traction Σ_{xx} at the edge of the plate. See England and McKenzie 1982 for instance.

It appears that the reviewer is incorrect. The lithospheric buoyancy forces in the overriding plate do not balance the mantle-flow-induced basal tractions, because the lithospheric buoyancy forces also promote extension through the positively buoyant overriding plate crust, just like the basal tractions promote extension during the early stage of subduction, which is due to the positive trench-normal horizontal gradient of the average horizontal trench-normal shear rate (Fig. 6a4). The horizontal compressive deviatoric stress σ_{xx} at the edge of the overriding plate due to shearing at the subduction zone interface is very localized and confined to a zone within ~50 km of the subduction zone interface (see for example Fig. 7b, stress pattern at 31.7 Myr), and so does not affect the backarc region, as is indeed evident from the tensional deviatoric stress in the backarc region (see Fig. 7a, red curve, and Fig. 7b, stress pattern at 31.7 Myr).

- L. 91-92: periodic folding: how relevant is that, in fact? Perhaps you could skip this part?

The periodic slab folding is important as it provides an explanation for the periodicity in the subducting plate velocity and trench velocity (Fig. 4a), the periodicity in the slab dip angle changes (Figs. 9a,b) and the periodic/episodic deformation in the overriding plate (Fig. 4a) (a point that reviewer 2 refers to as "intriguing").

- L. 96-98: the fact that the timing is OK can't really be regarded as a validating proof. Timing scales with viscosity and can be tuned accordingly (within a certain limit).

The text does not mention that it is a validating proof. The text mentions that there is good correspondence and significant overlap between model and nature. Besides, viscosity cannot be changed very much to scale time, as the velocities would then not be realistic. Also, the dimensionalized viscosity values are close to values suggested for nature (see text in revised Methods section). In any case, the text has been modified slightly to remove a potential suggestion that the timing is a validating proof.

- L. 111-114: it would be nice to see the streamlines on fig. 3.

Streamlines are o.k. in case the target audience is limited to geodynamicists and fluid dynamicists. However, considering the target audience is much larger here, including geologists, volcanologists, petrologists, structural geologists, tectonicists, geochemists, geophysicists, and geodesists, velocity vectors are preferred as they are intuitively much more understandable to such a diverse audience, given that the vector itself indicated the direction of flow and the size of the vector gives an indication of the velocity magnitude.

- L. 123-127: in our 2012 paper (Husson et al., EPSL), we also show a latitudinal variation of the mantle drag, that increases towards the Central Andes from both ends. This might actually be viewed as supporting this results.

Yes, interesting point. This is now mentioned in the section “Mechanism of overriding plate deformation”.

- L. 144-146: Trench migration is one of the favorite diagnostic tool to test plate tectonic models. The current models seems to be at odds with observations, from fig. 4e, but this is not quite clear. How do they compare to, for instance, Ren et al.?

Yes, good point. However, the current geodynamic model is actually quite consistent with observations. Average trench retreat velocities as deduced from Fig. 4e give a minimum of 1.1 cm/yr for the centre and a maximum of 1.3 cm/yr near the edges over a 182.6 Myr period. Such rates are of the same order of magnitude as average trench retreat rates for the last ~200 Myr ($\sim 1.8 \pm 0.3$ cm/yr in the centre and $\sim 1.9-2.1 \pm 0.3$ cm/yr near the edges) and are comparable to observed present-day rates of trench migration (-0.7 to 1.6 cm/yr) [Schellart et al., 2007], which also show the pattern of slower trench retreat rates in the centre compared to the edges.

I am not familiar with the work “Ren et al.”.

In any case, part of the text from above has been included in the section “Evolution of trench curvature and trench migration”.

Regards,
Laurent Husson

Reviewers' comments:

Reviewer #1 (Remarks to the Author):

Reviewer Number 1 Post-Revision Comments:

I am satisfied with the revised manuscript, and recommend the paper for publication. The revision places the results in the broader context of previous work and more specifically identifies model limitations. I have no major comments on the revision.

The very minor comments include:

(1) with the additional text placing the results in comparison to more detailed observations, move the interpretation into separate paragraphs from the results, i.e., new paragraph at lines 103 and lines 143;

(2) this is still only one model, which inherently contains many assumptions and limitations, thus in lines 179-180, change "is a consequence of" to "can be explained by".

Best regards,

Margarete Jadamec (Reviewer 1)

Reviewer #2 (Remarks to the Author):

This is the second time I am reviewing the submitted manuscript from Schellart. My original comments on the overall contribution of the manuscript still stands. I have read both the reviewers critiques and authors response as well we the revised manuscript. I think the changes that Schellart has included make the manuscript stronger, easier to read and a solid contribution to Nature Communications. my only addition suggestions are some small wording changes to help readability (shorten sentences).

Lines 57-60 change to "or overriding plate velocity. Explanations for trench-parallel variation in Andean shortening and topography call on trench-parallel variations sediment thickness or subducting plate age."

Line 195 add a However. However, what is important to note is....

Line 227-234 Change to: "Jurassic-Cretaceous rocks record a range of anticlockwise rotation values ($\sim 5-70^\circ$) north of the Arica bend and clockwise rotations values ($0-50^\circ$) south of the bend. The directions of these rotations (anticlockwise in the north and clockwise in the south) are consistent with the oroclinal rotations observed in the model. However, maximum model rotations ($\pm 16^\circ$) are on the low side when compared to recorded measurements..... and/or partially due to local block rotations in nature."

Reviewer #3 (Remarks to the Author):

In this revised version, W. Schellart addresses most reviewers' comments, including mine. Mostly in a satisfactory manner, but I still have several issues.

- A recurrent aspect of the reviews was pointing at the novelty of this work with respect to earlier

works, and this has not seriously be treated. In particular, with respect to the work of Faccenna et al. (2013). I find it unethical to read, at the very beginning of the text (l. 28-30), presented as the main result of the paper and as a novelty, that « [...] upper mantle subduction causes backarc extension, while whole (upper+lower) mantle subduction drives Andean orogeny », which is very close to what Faccenna et al (2013) state in the abstract « [...] “slab pull” type, where subduction is mainly confined to the upper mantle, and rollback trench motion lead to moderately thick crustal stacks[...]. Second, those of “slab suction” type, where whole-mantle convection cells [...] lead to the more extreme expressions of orogeny, such as the largely thickened crust and high plateaus of present-day Tibet and the Altiplano. ».

The fact that the current study adds a physical test to this idea is certainly interesting, but the original idea shall not be thought as an outcome of the current study. This comment also holds for more marginal earlier contributions. Please give justice to earlier works.

- One of the principal comment that we (reviewers) had in mind was that the paper was written in an over-optimistic way that was possibly irritating for the reader: misfits were mostly hidden and fits were systematically considered as validating proofs. Misfits are now better discussed. However, successful matches between model and data are still considered in the text as a proof of the soundness of the model. In fact, given the uncertainties in the parametrization (in the viscosity, for instance, as discussed in the rebuttal letter by the author), the magnitude of the metrics and the timing of evolution could vary by orders of magnitude. This is clearly acknowledged in the rebuttal letter by the author, but not that much in the article itself. I don't think it would downgrade the paper to present things as they are.

- No, free slip lateral BC is not a trivial choice. Even if it is commonly used and difficult to do otherwise. It is a strong choice because neither plates nor convection cells on Earth are closed systems: they are influenced by neighboring systems. Zero horizontal velocity at the BC means stresses adjust to make free slip hold throughout. As W. Schellart says, the model box has indeed a very large extent, and therefore BC are in the far field with respect to the Andes in the model. As such, they perhaps do not need to quickly adjust through time, and it likely implies that Σ_{xx} along the vertical BC are close to zero throughout. This is not ok, a priori. I don't think there is any argument to say that Σ_{xx} is zero in the far field. Instead, it has good reasons to be anything else. The fact that the « Pangean cell » competes with the « Panthalassan cell » through time (e.g. Collins, 2003; or Davaille et al., 2005) certainly reveals that the lateral boundaries are not stress free.

I understand that very many models make the same sort of assumption and feel that it is a fine justification of their choice. I simply contend that it is a default assumption that is not as good as one could think. « Minimiz[ing] the influence of these boundaries on the flow in the system » is a strong choice, and there is no obvious reason for that.

I therefore maintain that the justification in the text for the BC is thus not suitable from a general standpoint. Regional models include this bias, which I believe is not trivial, for the above reasons. To put it bluntly, I believe that this choice shall not be presented as a satisfying decision but instead as a mere attempt to test the intrinsic processes in the center of the box. That is what it is, and I suggest to write is as such.

- On the force balance: I maintain that, unlike in W. Schellart's answer, lithospheric buoyancy forces contribute to balance the forces. They are not outside the system. But I might not understand W. Schellart's answer properly, in fact. Stresses are continuous and do not simply vanish. But this is probably outside the scope of this paper.

Response to reviewers

Please find below my responses (in Arial blue) to the original comments from the reviewers (in Times black).

Reviewer 1:

Reviewer #1 (Remarks to the Author):

Reviewer Number 1 Post-Revision Comments:

I am satisfied with the revised manuscript, and recommend the paper for publication. The revision places the results in the broader context of previous work and more specifically identifies model limitations. I have no major comments on the revision.

The very minor comments include:

(1) with the additional text placing the results in comparison to more detailed observations, move the interpretation into separate paragraphs from the results, i.e., new paragraph at lines 103 and lines 143; **The new paragraph breaks have been added.**

(2) this is still only one model, which inherently contains many assumptions and limitations, thus in lines 179-180, change "is a consequence of" to "can be explained by".

Yes, agreed. The text has been changed according to the suggestion from the reviewer.

Best regards,

Margarete Jadamec (Reviewer 1)

Reviewer 2:

Reviewer #2 (Remarks to the Author):

This is the second time I am reviewing the submitted manuscript from Schellart. My original comments on the overall contribution of the manuscript still stands. I have read both the reviewers critiques and authors response as well we the revised manuscript. I think the changes that Schellart has included make the manuscript stronger, easier to read and a solid contribution to Nature Communications. my only addition suggestions are some small wording changes to help readability (shorten sentences).

Lines 57-60 change to "or overriding plate velocity. Explanations for trench-parallel variation in Andean shortening and topography call on trench-parallel variations sediment thickness or subducting plate age."

The text has been modified according to the suggestion from the reviewer.

Line 195 add a However. However, what is important to note is....

"However" has been added.

Line 227-234 Change to: "Jurassic-Cretaceous rocks record a range of anticlockwise rotation values (~5-70°) north of the Arica bend and clockwise rotations values (0-50°) south of the bend. The directions of these rotations (anticlockwise in the north and clockwise in the south) are consistent with the oroclinal rotations observed in the model. However, maximum model rotations (+/-) 16° are on the low side when compared to recorded measurements..... and/or partially due to local block rotations in nature."

The text has been modified according to the suggestion from reviewer 2 above.

Reviewer 3:

Reviewer #3 (Remarks to the Author):

In this revised version, W. Schellart addresses most reviewers' comments, including mine. Mostly in a satisfactory manner, but I still have several issues.

- A recurrent aspect of the reviews was pointing at the novelty of this work with respect to earlier works, and this has not seriously be treated.

By using the plural form "reviews" above, reviewer 3 (L. Husson) implies that the other two reviewers (1 & 2) also questioned the novelty of this paper, which they did not.

In particular, with respect to the work of Faccenna et al. (2013). I find it unethical to read, at the very beginning of the text (l. 28-30), presented as the main result of the paper and as a novelty, that « [...] upper mantle subduction causes backarc extension, while whole (upper+lower) mantle subduction drives Andean orogeny »,

I find the partial quotation from above misleading. The entire sentence reads: "*Here I present a four-dimensional buoyancy-driven whole-mantle subduction model implying that the ~200 Myr geological history can be attributed to sinking of a wide slab into a layered mantle, where upper mantle subduction causes backarc extension, while whole (upper+lower) mantle subduction drives Andean orogeny.*" There are many important aspects to this sentence:

1. The geodynamic subduction model is buoyancy-driven
2. The model is 4D and extends for the entire mantle depth
3. The model simulates a very wide slab.
4. The model simulates a very long timescale (~200 Myr).
5. The model can explain important aspects of the 200 Myr geological evolution in the Andean region, which are attributed to subduction of a wide slab into a layered mantle.
6. The model implies that upper mantle subduction of a wide slab drives backarc extension.
7. The model implies that whole mantle subduction of a wide slab drives Andean orogeny.

Almost all of these points are not covered in Faccenna et al. [2013].

which is very close to what Faccenna et al (2013) state in the abstract « [...] "slab pull" type, where subduction is mainly confined to the upper mantle, and rollback trench motion lead to moderately thick crustal stacks[...]. Second, those of "slab suction" type, where whole-mantle convection cells [...] lead to the more extreme expressions of orogeny, such as the largely thickened crust and high plateaus of present-day Tibet and the Altiplano. ».

I do not see how my sentence is very similar to what Faccenna et al. [2013] say in the quotation above. They do not have a geodynamic subduction model, they do not simulate very long timescales, and most importantly, they do not attribute the history and the mountain building to subduction of a wide slab like the one subducting below South America, which I argue is crucial. On its own, whole mantle convection (due to deep subduction into the lower mantle) does not guarantee cordilleran orogeny (e.g. narrow Hellenic slab (~800 km) with subduction down to 1,500 km [e.g. van Hinsbergen et al., 2005], but Aegean backarc extension).

I would argue that only the last two points above (6 and 7) only partly overlap with the quotation from Faccenna et al. [2013] above and the rest of their paper. I say partly, because Faccenna et al. speak of "*moderately thick crustal stacks*" and "*slab pull - orogeny*" (their Fig. 7) for subduction confined to the upper mantle, while I speak of "*backarc extension*", which is very different considering that there is formation of a backarc basin rather than formation of an orogen. As for point 7, Faccenna et al. speak of "*whole mantle convection cells*" and "*Tibet and the Altiplano*", while I speak of "*whole (upper+lower) mantle subduction*" and "*Andean Orogeny*". And again, Faccenna et al. do not mention slab width as a crucial ingredient, which I mention throughout the manuscript as crucial for Andean orogeny.

The fact that the current study adds a physical test to this idea is certainly interesting, but the original idea shall not be thought as an outcome of the current study. This comment also holds for more

marginal earlier contributions. Please give justice to earlier works.

81 earlier works are cited and discussed in the manuscript. To emphasize the novel aspect of my work, that a wide slab is crucial for Andean orogeny (in combination with lower mantle subduction), I have modified the text in the abstract (L29-30) and results section (L184-188). The second revision now cites and discusses the work of Faccenna et al. [2013]. See in particular the last paragraph of the section "Mechanism of overriding plate deformation". I find the reviewer's statement: "*This comment also holds for more marginal earlier contributions.*" inappropriate. First, my earlier contributions are not the subject of review here. Second, to call my earlier contributions "*marginal*" is rather subjective and improper.

- One of the principal comment that we (reviewers) had in mind was that the paper was written in an over-optimistic way

With the words "we (reviewers)", reviewer L. Husson implies that the other two reviewers also argued that the paper was overly optimistic, which is not the case. Reviewer 1 (Jadamec) certainly never mentioned this in her review.

that was possibly irritating for the reader: misfits were mostly hidden and fits were systematically considered as validating proofs.

Husson implies as if it was my intention to hide misfits, which I entirely disagree with. As I wrote extensively in the earlier response letter, the Jurassic-E. Cretaceous extension in the Central Andes has never been quantified. Reviewer 2 (McQuarrie) provided convincing arguments that Jurassic-E. Cretaceous extension in the Central Andes was likely minor. And in the revised manuscript I accommodated these comments from McQuarrie.

Misfits are now better discussed. However, successful matches between model and data are still considered in the text as a proof of the soundness of the model.

Throughout the manuscript I now mostly say that model results are consistent with observations (or something similar). Furthermore, I have changed the text in a few more places to avoid "irritating" the reader (e.g. L142-143, L180-181, L211).

In fact, given the uncertainties in the parametrization (in the viscosity, for instance, as discussed in the rebuttal letter by the author), the magnitude of the metrics and the timing of evolution could vary by orders of magnitude. This is clearly acknowledged in the rebuttal letter by the author, but not that much in the article itself. I don't think it would downgrade the paper to present things as they are.

The uncertainty in viscosity in nature is discussed in the Methods section, second paragraph, and I have added a statement in the Methods section (second paragraph) that the scaling formulations indicate how uncertainty in mantle viscosity in nature translates in uncertainty in dimensionalized velocity.

- No, free slip lateral BC is not a trivial choice. Even if it is commonly used and difficult to do otherwise. It is a strong choice because neither plates nor convection cells on Earth are closed systems: they are influenced by neighboring systems. Zero horizontal velocity at the BC means stresses adjust to make free slip hold throughout. As W. Schellart says, the model box has indeed a very large extent, and therefore BC are in the far field with respect to the Andes in the model. As such, they perhaps do not need to quickly adjust through time, and it likely implies that σ_{xx} along the vertical BC are close to zero throughout. This is not ok, a priori. I don't think there is any argument to say that σ_{xx} is zero in the far field. Instead, it has good reasons to be anything else. The fact that the « Pangean cell » competes with the « Panthalassan cell » through time (e.g. Collins, 2003; or Davaille et al., 2005) certainly reveals that the lateral boundaries are not stress free.

Sure, in nature σ_{xx} in the far-field is not zero, as there are other systems and processes in the far-field that operate, e.g. subduction zones in the Western Pacific like Tonga-Kermadec and Mariana. I would argue that such far-field subduction zones have a negligible influence on the mountain building in the Andes, and that it is the South American subduction system that is predominantly driving Andean orogeny. One way to test this hypothesis is to build a regional subduction model that only includes the South American subduction zone (no other subduction zones or global mantle flow patterns) and then investigate if this subduction zone can, on its own, produce the first-order features of the Andes. In such a scenario, one wants

to minimize the influence of the lateral boundaries. The model indeed reproduces (to the first order) several important features of the Andes, e.g.: Andean orogeny, crustal thickening, northward and southward decrease in shortening and crustal thickening, Jurassic-E. Cretaceous backarc extension, the Bolivian orocline, westward motion of the South American plate and eastward migration of the Central Andean magmatic arc.

I understand that very many models make the same sort of assumption and feel that it is a fine justification of their choice. I simply contend that it is a default assumption that is not as good as one could think. « Minimiz[ing] the influence of these boundaries on the flow in the system » is a strong choice, and there is no obvious reason for that.

If the main goal of the regional model is to investigate the effect of the local subduction zone (so excluding far-field effects), then I (and likely many others) think this choice is justified. I could add that Yamato et al. [GRL 2009] use free-slip boundary conditions as well in their regional subduction model.

I therefore maintain that the justification in the text for the BC is thus not suitable from a general standpoint. Regional models include this bias, which I believe is not trivial, for the above reasons. To put it bluntly, I believe that this choice shall not be presented as a satisfying decision but instead as a mere attempt to test the intrinsic processes in the center of the box. That is what it is, and I suggest to write is as such.

In the Methods section I have added that it is a regional subduction model that investigates the subduction process of the South American subduction zone in isolation (excluding far-field effects related to, e.g. other subduction zones or global mantle flow), and therefore the boundary conditions (free-slip) and size of the box were chosen as such that their effect on the subduction process in the model was minimized.

- On the force balance: I maintain that, unlike in W. Schellart's answer, lithospheric buoyancy forces contribute to balance the forces. They are not outside the system. But I might not understand W. Schellart's answer properly, in fact. Stresses are continuous and do not simply vanish. But this is probably outside the scope of this paper.

The numerical geodynamic model presented in the manuscript includes lithospheric buoyancy forces, e.g. the light continental crust in the overriding plate. Furthermore, these lithospheric buoyancy forces change in the model due to the evolution of the continental crustal thickness. So I do not see a problem here. I would further add that lithospheric buoyancy forces can contribute to the force balance at plate boundaries in case they are not encapsulated within a (strong) tectonic plate, and I agree with the reviewer that lithospheric buoyancy forces are not outside the system. So again, I do not see a problem.

REVIEWERS' COMMENTS:

Reviewer #3 (Remarks to the Author):

I feel that this revised version is more cautious on several aspects that I mentioned and has improved. It is probably time to move on with this paper, and get it published.

Regards,
Laurent Husson

Response to reviewers

Please find below my responses (in Arial blue) to the original comments from the reviewers (in Times black).

Reviewer 3:

Reviewer #3 (Remarks to the Author):

I feel that this revised version is more cautious on several aspects that I mentioned and has improved. It is probably time to move on with this paper, and get it published.

Regards,

Laurent Husson

No response is required.